# How does over-squashing affect the power of GNNs?

**Francesco Di Giovanni**[*]                                                   *francesco.di.giovanni@cs.ox.ac.uk*
*University of Oxford*

**T. Konstantin Rusch**[*]                                                               *tkrusch@mit.edu*
*Massachusetts Institute of Technology*

**Michael M. Bronstein**
*University of Oxford*

**Andreea Deac**
*Université de Montréal*

**Marc Lackenby**
*University of Oxford*

**Siddhartha Mishra**
*ETH Zürich*

**Petar Veličković**
*Google DeepMind*

**Reviewed on OpenReview:** *https://openreview.net/forum?id=KJRoQvRWNs*

## Abstract

Graph Neural Networks (GNNs) are the state-of-the-art model for machine learning on graph-structured data. The most popular class of GNNs operate by exchanging information between adjacent nodes, and are known as Message Passing Neural Networks (MPNNs). While understanding the expressive power of MPNNs is a key question, existing results typically consider settings with uninformative node features. In this paper, we provide a rigorous analysis to determine which function classes of node features can be learned by an MPNN of a given capacity. We do so by measuring the level of *pairwise interactions* between nodes that MPNNs allow for. This measure provides a novel quantitative characterization of the so-called over-squashing effect, which is observed to occur when a large volume of messages is aggregated into fixed-size vectors. Using our measure, we prove that, to guarantee sufficient communication between pairs of nodes, the capacity of the MPNN must be large enough, depending on properties of the input graph structure, such as commute times. For many relevant scenarios, our analysis results in impossibility statements in practice, showing that *over-squashing hinders the expressive power of MPNNs*. Our theory also holds for geometric graphs and hence extends to equivariant MPNNs on point clouds. We validate our analysis through extensive controlled experiments and ablation studies.

## 1 Introduction

Graphs describe the relational structure for a large variety of natural and artificial systems, making learning on graphs imperative in many contexts (Veličković, 2023; DeZoort et al., 2023; Williamson, 2023). To this end, Graph Neural Networks (GNNs) (Gori et al., 2005; Scarselli et al., 2008) have emerged as a widely popular framework for graph machine learning, with plentiful success stories in science (Bapst et al., 2020;

---

[*] Equal contribution.

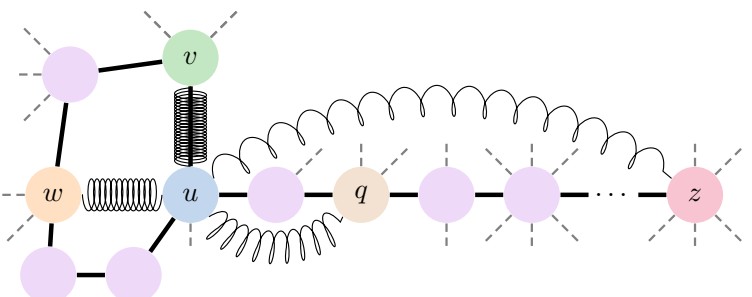

Figure 1: We study the power of MPNNs in terms of the mixing they induce among features and show that this is affected by the model (via norm of the weights and depth) and the graph topology (via commute times). For the given graph, the MPNN learns stronger mixing (tight springs) for nodes $v, u$ and $u, w$ since their commute time is small, while nodes $u, q$ and $u, z$, with high commute-time, have weak mixing (loose springs). We characterize over-squashing as the inverse of the mixing induced by an MPNN and hence relate it to its power. In fact, the MPNN might require an impractical depth to solve tasks on the given graph that depend on high-mixing of features assigned to $u, z$.

Davies et al., 2021; Blundell et al., 2021; Stokes et al., 2020; Liu et al., 2023) and technology (Monti et al., 2019; Mirhoseini et al., 2021; Derrow-Pinion et al., 2021).

Given an underlying graph and features, defined on its nodes (and edges), as inputs, a GNN learns parametric functions from data. Due to the ubiquity of GNNs, characterizing their **expressive power**, i.e., which class of functions a GNN is able to learn, is a problem of great interest. In this context, most available results in literature on the universality of GNNs pertain to impractical higher-order tensors (Maron et al., 2019; Keriven & Peyré, 2019) or unique node identifiers that may break the symmetries of the problem (Loukas, 2020). In particular, these results do not necessarily apply to Message Passing Neural Networks (MPNNs) (Gilmer et al., 2017), which have emerged as the most popular class of GNN models in recent years. Concerning expressivity results for MPNNs, the most general available characterization is due to Xu et al. (2019) and Morris et al. (2019), who proved that MPNNs are, at most, as powerful as the Weisfeiler-Leman graph isomorphism test (Weisfeiler & Leman, 1968) in distinguishing graphs *without any features*. This brings us to an important question:

> Which classes of functions can MPNNs of a given capacity learn, *if node features are specified*?

Razin et al. (2022) address this question by characterizing the separation rank of MPNNs; however, their analysis only covers *unconventional* architectures that do not correspond to MPNN models used in practice. In contrast, Alon & Yahav (2021) investigate this question *empirically*, by observing that MPNNs fail to solve tasks which involve *long-range interactions* among nodes. This limitation was ascribed to a phenomenon termed as **over-squashing**, which loosely entails messages being 'squashed' into fixed-size vectors when the receptive field of a node grows too fast. This effect was formalized in Topping et al. (2022); Di Giovanni et al. (2023); Black et al. (2023), who showed that the Jacobian of the nodes features is affected by topological properties of the graph, such as curvature and effective resistance. However, all the aforementioned papers ignore the specifics of the *task* at hand, i.e., the underlying *function* that the MPNN seeks to learn, leading us to the following question:

> How does *over-squashing affect the expressive power* of MPNNs? Can we *measure* it?

**What about geometric graphs?** In many scientific applications, data come as graphs embedded in Euclidean space. Since popular architectures resort to the message-passing paradigm (Gasteiger et al., 2021; Dauparas et al., 2022; Batzner et al., 2022), the expressive power of such models has been rephrased in the language of the WL test, once extended to account for the extra geometric information (Joshi et al., 2023). Nonetheless, the questions raised above are even more pressing for these tasks, where the graph is typically derived from a point cloud using a cutoff radius, while the features also contain information about the positions in 3D space. In fact, for such problems where the features may carry more valuable information

than the 2D graph structure, we argue that proposing new ways to assess the power of message-passing other than (variants of) the WL test, is crucial. To this aim, in our paper we study generic message-passing equations with no assumptions on the nature of the features, meaning that they may also **include additional positional information** if the dataset is a point cloud embedded in Euclidean space.

**Contributions.** Our main goal is to show how over-squashing can be understood as the *misalignment between the task and the graph-topology*, thereby *limiting the classes of functions that MPNNs of practical size can learn* (see Figure 1). We start by measuring the extent to which an MPNN allows any pair of nodes in the graph to interact (via **mixing** their features). With this measure as a tool, we characterize which functions of node features can be learned by an MPNN and how the model architecture and parameters, as well as the topology of the graph, affect the expressive power. More concretely,

- We introduce a novel metric of expressive power that is based on the Hessian of the function learned by an MPNN and measures the ability of a model to mix features associated with different nodes. This approach complements the current repertoire of tools at disposal to assess the power of GNNs, and is of interest independent of the over-squashing phenomenon.

- We then prove upper bounds on the Hessian of functions learned by MPNNs, hence deriving limitations on the power of MPNNs to mix features (i.e., model interactions) according to the novel metric mentioned above. As far as we know, this is the first theoretical result stating limitations of MPNNs to learn functions *and their derivatives.*

- We characterize over-squashing as the reciprocal of the maximal mixing that MPNNs can induce among nodes: *the higher this measure, the smaller the class of functions MPNNs can learn.* Differently from the WL-test, which depends on the separation of node representations based on the computational trees, our metric quantifies the ability of an MPNN to mix information associated with different nodes and the cost required to do so in terms of size of the weights and depth.

- We prove that the weights and depth must be sufficiently large – depending on the topology – to ensure mixing. For some tasks, *the depth must exceed the highest commute time on the graph*, resulting in *impossibility* statements. Our results show that MPNNs of practical size, fail to learn functions with strong mixing among nodes at high commute time.

- We illustrate our theoretical results with controlled experiments that verify our analysis, by highlighting the impact of the architecture (depth), of the topology (commute time), and of the underlying task (the level of mixing required).

Thus, we present a novel paradigm for deriving rigorous results in relating two key notions in graph machine learning, that of GNN expressive power and over-squashing.

## 2 The Message-Passing paradigm

**Definitions on graphs.** We denote a graph by $\mathsf{G} = (\mathsf{V}, \mathsf{E})$, where $\mathsf{V}$ is the set of $n$ nodes while $\mathsf{E}$ are the edges. We assume that $\mathsf{G}$ is *undirected, connected* and non-bipartite and define the $n \times n$ adjacency matrix $\mathbf{A}$ as $A_{vu} = 1$ if $(v, u) \in \mathsf{E}$ and zero otherwise. We let $\mathbf{D}$ be the diagonal degree matrix with $D_{vv} := d_v$ and use $d_{\max}$ and $d_{\min}$ to denote the maximal and minimal degrees. Since we are interested in the over-squashing phenomenon, which affects the propagation of information, we need to quantify distances on $\mathsf{G}$. We let $d_{\mathsf{G}}(v, u)$ be the length of the shortest path connecting nodes $v$ and $u$ (geodesic distance). While $d_{\mathsf{G}}$ describes how far two nodes $u, v$ are in $\mathsf{G}$, it does not account for how many different routes they can use to communicate. In fact, we will see below that the over-squashing of nodes $v, u$ and, more generally, the mixing induced by MPNNs among the features associated with $v, u$, can be better quantified by their **commute time** $\tau(v, u)$, equal to the expected number of steps for a random walk to start at $v$, reach $u$, and then come back to $v$.

**The MPNN-class.** For most problems, graphs are equipped with features $\{\mathbf{x}_v\}_{v \in \mathsf{V}} \subset \mathbb{R}^d$, whose matrix representation is $\mathbf{X} \in \mathbb{R}^{n \times d}$. To study the interactions induced by a GNN among pairs of features, we focus on *graph-level* tasks – in Section E of the Appendix, we extend the discussion and our main theoretical results

to *node-level* tasks. The goal then is to predict a function $\mathbf{X} \mapsto y_{\mathsf{G}}(\mathbf{X})$, where we assume that the graph $\mathsf{G}$ is fixed and thus $y_{\mathsf{G}} : \mathbb{R}^{n \times d} \to \mathbb{R}$ is a function of the node features. MPNNs define a family of parametric functions through iterative local updates of the node features: the feature of node $v$ at layer $t$ is derived as

$$\mathbf{h}_v^{(t)} = f^{(t)}\left(\mathbf{h}_v^{(t-1)}, g^{(t)}\left(\{\!\{\mathbf{h}_u^{(t-1)}, \; (v,u) \in \mathsf{E}\}\!\}\right)\right), \quad \mathbf{h}_v^{(0)} = \mathbf{x}_v \tag{1}$$

where $f^{(t)}, g^{(t)}$ are learnable functions, $g^{(t)}$ is invariant to permutations and $\{\!\{\cdot\}\!\}$ denotes a multiset – meaning terms can be repeated. Specifically, we study a class of MPNNs of the following form,

$$\mathbf{h}_v^{(t)} = \sigma\left(\mathbf{\Omega}^{(t)}\mathbf{h}_v^{(t-1)} + \mathbf{W}^{(t)}\sum_u \mathsf{A}_{vu}\psi^{(t)}(\mathbf{h}_v^{(t-1)}, \mathbf{h}_u^{(t-1)})\right), \quad \mathbf{h}_v^{(0)} = \mathbf{x}_v, \tag{2}$$

where $\sigma$ acts pointwise, $\mathbf{\Omega}^{(t)}, \mathbf{W}^{(t)} \in \mathbb{R}^{d \times d}$ are weight matrices, $\mathbf{A} \in \mathbb{R}^{n \times n}$ is *any* matrix satisfying $\mathsf{A}_{vu} > 0$ if $(v, u) \in \mathsf{E}$ and zero otherwise – $\mathbf{A}$ is typically some (normalized) version of the adjacency matrix $\mathbf{A}$ – and $\psi^{(t)}$ is a learnable message function. The layer-update in (2) includes common MPNN-models such as GCN (Kipf & Welling, 2017), SAGE (Hamilton et al., 2017), GIN (Xu et al., 2019), and GatedGCN (Bresson & Laurent, 2017). As commented extensively in Section 6, this is the most general class of MPNN equations studied thus far in theoretical works on over-squashing; unless otherwise stated, all our considerations and analysis apply to MPNNs as in (2). For graph-level tasks, a permutation-invariant readout READ is required – usually MAX, MEAN, or SUM. We define the graph-level function computed by the MPNN after $m$ layers to be

$$y_{\mathsf{G}}^{(m)}(\mathbf{X}) = \boldsymbol{\theta}^\top \mathsf{READ}(\{\!\{\mathbf{h}_v^{(m)}\}\!\}), \tag{3}$$

for some learnable $\boldsymbol{\theta} \in \mathbb{R}^d$. While a more complex non-linear readout can be considered, we restrict to a single linear-layer since we are interested in the mixing among variables induced by the MPNN itself through the topology (and not in readout, independently of the graph-structure).

**MPNNs on geometric graphs.** We note that (2) also describes a class of generic, *equivariant* MPNNs over a point cloud embedded in Euclidean space, once the matrix $\mathbf{A}$ is intended to encode the pairs of points that exchange information across each layer—typically creating a graph $\mathsf{G}$ by drawing a weighted edge between any pair of points whose distance is smaller than a cutoff radius. In fact, throughout our analysis we have no restriction on the type of features $\mathbf{h}_v$, that can also contain the position of a node in 3D space. Accordingly, our theoretical results below hold for MPNNs on both 2D and 3D data, since they pertain to how the message passing paradigm models pairwise interactions among different points (nodes). To streamline the presentation, we often omit to specify that the features can also encode 3D information (when available).

## 3 On the mixing induced by Message Passing Neural Networks

As one of the main contributions of this paper, we propose a new framework for characterizing the expressive power of MPNNs by *estimating the amount of mixing they induce among pairs of features* $\mathbf{x}_v$ and $\mathbf{x}_u$, corresponding to the nodes $v$ and $u \in \mathsf{V}$. To motivate our definition, fix the underlying graph $\mathsf{G}$, let $y_{\mathsf{G}}$ be the ground-truth function to be learned, and suppose, for simplicity, that the node features $\{x_i\}$ are all scalar. If $y_{\mathsf{G}}$ is a smooth function, then we can take the Taylor expansion of $y_{\mathsf{G}}$ at any point $\bar{\mathbf{x}} = (\bar{x}_1, \ldots, \bar{x}_n)$ and obtain a polynomial in the variables $(x_1, \ldots, x_n)$, up to higher-order corrections. The *mixing* induced by $y_{\mathsf{G}}$ on the features $x_v, x_u$ can then be expressed in terms of *mixed* product monomials of the form $x_v x_u$, and the powers thereof. The lowest-degree mixed monomials of this form are multiplied by the Hessian (i.e. the second-order derivatives) of $y_{\mathsf{G}}$. Accordingly, we can take the entries $v, u$ of the Hessian of $y_{\mathsf{G}}$ as the simplest **measure of pairwise mixing** induced by $y_{\mathsf{G}}$ over the nodes $v, u$. In fact, to make things more concrete, note that if $y_{\mathsf{G}}(\mathbf{x}) = \phi_0(x_v) + \phi_1(x_u)$, then the mixing (Hessian) would always be zero since the variables are fully separable; conversely, if $y_{\mathsf{G}}(\mathbf{x}) = \phi(\mathbf{x}_v^\top \mathbf{x}_u)$, then the amplitude of the mixing (Hessian) depends on how *nonlinear* $\phi$ is. The same reasoning applies to higher-dimensional node features too:

**Definition 3.1.** *For a twice differentiable graph-function* $y_{\mathsf{G}}$ *of node features* $\{\mathbf{x}_i\}$, *the* **maximal mixing** *induced by* $y_{\mathsf{G}}$ *among the features* $\mathbf{x}_v$ *and* $\mathbf{x}_u$ *associated with nodes* $v, u$ *is*

$$\mathsf{mix}_{y_{\mathsf{G}}}(v, u) = \max_{\mathbf{x}_i} \max_{1 \le \alpha, \beta \le d} \left|\frac{\partial^2 y_{\mathsf{G}}(\mathbf{X})}{\partial x_v^\alpha \partial x_u^\beta}\right|. \tag{4}$$

We note that the first maximum is taken over all input features – considering that the Hessian of $y_G$ is itself a function of the node features over the graph – while the second maximum is taken over all entries $\alpha, \beta$ of the $d$-dimensional node features $\mathbf{x}_v$ and $\mathbf{x}_u$; it is straightforward to adapt the results below to alternative definitions based on different norms of the Hessian.

**Problem statement.** Our goal is to study the expressive power of MPNNs in terms of the (maximal) mixing they can generate among nodes $v, u$. *A low value of mixing implies that the MPNN cannot learn functions $y_G$ that require high mixing of the features associated with $v, u$ and hence they cannot model 'product'-type interactions* as per our explanation based on the Taylor expansion. We investigate how weights and depth on the one side, and the graph topology on the other, affect the mixing of an MPNN.

**The requirement of smoothness.** In many applications, especially when deploying neural network models to solve partial differential equations, the predictions need to be sufficiently regular (smooth), which motivates the adoption of smooth non-linear activations (Hornik et al., 1990; Brandstetter et al., 2021; Equer et al., 2023). Our analysis below follows this paradigm and holds for all activations $\sigma$ that are (at least) twice differentiable, including smoother versions of ReLU such as the GELU function.

## 3.1 Pairwise mixing induced by MPNNs

In this Section, our goal is to derive an upper bound on the maximal mixing induced by MPNNs, as defined above, over the features associated with pairs of nodes $v, u$. To *motivate the structure of this bound*, we consider the simple yet illustrative setting of an MPNN as in (2) with scalar features, weights $\omega, \mathsf{w} > 0$ and a linear message function of the form $\psi(x, y) = c_1 x + c_2 y$, for some learnable constants $c_1, c_2$. In this case, the layer-update (2) takes the very simple form,

$$h_v^{(t)} = \sigma(\mathsf{w}(\mathbf{S}h^{(t-1)})_v), \quad \mathbf{S} := \frac{\omega}{\mathsf{w}}\mathbf{I} + c_1 \mathrm{diag}(\mathbf{A1}) + c_2 \mathbf{A} \in \mathbb{R}^{n \times n}, \tag{5}$$

where $\mathbf{1} \in \mathbb{R}^n$ is the vector of ones. Hence, the operator $\mathsf{w}\mathbf{S}$ governs the flow of information from layer $t-1$ to layer $t$ – once we factor out the derivatives of $\sigma$ – and the $k$-power of this matrix $(\mathsf{w}\mathbf{S})^k$ determines the propagation of information on the graph over $k$ layers, i.e. over walks of length $k$.

Suppose that nodes $v, u$ are at distance $r$ on the graph. If this MPNN has depth $m$, to have any mixing among the features of $v$ and $u$, there has to be a node $i$ that receives information from both $v$ and $u$, which only occurs when $m \geq r/2$. If $m = r/2$, then we can roughly estimate the mixing at node $i$ as $((\mathsf{w}\mathbf{S})^m)_{iv}((\mathsf{w}\mathbf{S})^m)_{iu}$. When $m > r/2$, information from $v$ and $u$ can arrive at $i$ after being processed at an intermediate node $j$; in fact, this node $j$ first collects information from both $v$ and $u$ over a walk of length $m - k$, and then shares it with node $i$ over a walk of length $k$. We then obtain terms of the form $\sum_k \sum_j ((\mathsf{w}\mathbf{S})^{m-k})_{jv}((\mathsf{w}\mathbf{S})^k)_{ij}((\mathsf{w}\mathbf{S})^{m-k})_{ju}$, since we are *summing over all possible ways of aggregating information from $v$ and $u$ across intermediate nodes $j$, before mixing it at $i$*.

A similar argument also works in the general case of (2), once we account for bounds on the non-linear activation function $\sigma$ by $c_\sigma = \max\{|\sigma'|, |\sigma''|\}$, and we choose $\omega, \mathsf{w}, c_1, c_2$ satisfying

$$\|\mathbf{\Omega}^{(t)}\| \leq \omega, \ \|\mathbf{W}^{(t)}\| \leq \mathsf{w}, \ \|\nabla_i \psi^{(t)}\| \leq c_i,$$

for $i = 1, 2$, where $\nabla_i \psi$ is the Jacobian of $\psi$ with respect to the $i$-th variable, and $\|\cdot\|$ is the *operator norm* of a matrix. We note that for trained MPNNs the weights would be finite and bounded, so our assumption is mild. For models with linear $\psi$, such as GCN, SAGE or GIN, these constants will suffice in deriving the upper bound on the mixing. However, in the more general case of non-linear message functions $\psi$, which for example includes GatedGCN, we also need to account for the term $\mathbf{Q}_k$, defined below, which arises due to the Hessian of $\psi$ when taking second-order derivatives of the MPNN (2): given $\mathbf{S}$ in (5), we set

$$\mathbf{P}_k := (\mathbf{S}^{m-k-1})^\top \mathrm{diag}(\mathbf{1}^\top \mathbf{S}^k)(\mathbf{AS}^{m-k-1})$$
$$\mathbf{Q}_k := \mathbf{P}_k + \mathbf{P}_k^\top + (\mathbf{S}^{m-k-1})^\top \mathrm{diag}(\mathbf{1}^\top \mathbf{S}^k(\mathrm{diag}(\mathbf{A1}) + \mathbf{A}))\mathbf{S}^{m-k-1}. \tag{6}$$

We assume that the Hessian of $\psi$ is bounded as $\|\nabla^2 \psi^{(t)}\| \leq c^{(2)}$. Recall that $y_G^{(m)}$ is the MPNN-prediction (3) and that $\mathrm{mix}_{y_G^{(m)}}(v, u)$ is its maximal mixing of nodes $v, u$ as per Definition 3.1.

**Theorem 3.2.** *Consider an* MPNN *of depth $m$ as in* (2)*, where $\sigma$ and $\psi^{(t)}$ are $\mathcal{C}^2$ functions and we denote the bounds on their derivatives and on the norm of the weights as above. Let $\mathsf{S}$ and $\mathbf{Q}_k$ be defined as in* (5) *and* (6)*, respectively. If the readout is* MAX, MEAN *or* SUM *and $\boldsymbol{\theta}$ in* (3) *has unit norm, then the mixing* $\mathsf{mix}_{y_{\mathsf{G}}^{(m)}}(v,u)$ *induced by the* MPNN *over the features of nodes $v, u$ satisfies*

$$\mathsf{mix}_{y_{\mathsf{G}}^{(m)}}(v, u) \leq \sum_{k=0}^{m-1} (c_\sigma \mathsf{w})^{2m-k-1} \left( \mathsf{w}(\mathsf{S}^{m-k})^\top \mathrm{diag}(\mathbf{1}^\top \mathsf{S}^k) \mathsf{S}^{m-k} + c^{(2)} \mathbf{Q}_k \right)_{vu}. \tag{7}$$

Theorem 3.2 shows *how* the mixing induced by an MPNN depends on the model (via regularity of $\sigma$, norm of the weights $\mathsf{w}$, and depth $m$) and on the graph-topology (via the powers of $\mathbf{A}$, which enters the definition of $\mathsf{S}$ in (5)). Despite its generality, from (7) it is as yet unclear which specific properties of the underlying graph affect the mixing the most, and how weights and depth of the MPNN can compensate for it. Therefore, our goal now is to expand (7) and relate it to known quantities on the graph and show how this can be used to characterize the phenomenon of over-squashing. First, we introduce a notion of *capacity* of an MPNN in the spirit of Loukas (2020).

**The capacity of an MPNN.** For simplicity, we **assume** that $c_\sigma = 1$, since this is satisfied by most commonly used non-linear activations – it is straightforward to extend the analysis to arbitrary $c_\sigma$.

**Definition 3.3.** *Given an* MPNN *with $m$ layers and $\mathsf{w}$ the maximal operator norm of the weights, we say that the pair $(m, \mathsf{w})$ represents the* **capacity** *of the* MPNN*.*

A larger capacity, by increasing either $m$ or $\mathsf{w}$, or both, heuristically implies that the MPNN has more expressive power and can hence induce larger mixing among the nodes $v, u$. Accordingly, given $v, u$, we formulate the problem of expressivity as: *what is the capacity required to induce enough mixing $\mathsf{mix}_{y_{\mathsf{G}}}(v, u)$?*

**Studying expressivity through derivatives.** In applications to physics and PDEs, we may often need the neural-network to also match derivatives of a target function (Hornik et al., 1990). Theorem 3.2 shows that the second-order derivatives of MPNN predictions as in (2), **cannot** approximate second-order derivatives of graph-functions $y_{\mathsf{G}}$ whose associated mixing is larger than the right hand side of (7). To the best of our knowledge, this is the first theoretical analysis on the limitations of MPNNs to approximate classes of functions and the derivatives thereof.

> **Message of the Section:** *We can estimate the* **nonlinear, joint** *dependence of a smooth graph function on pairs of node features in terms of its second-order derivatives. We refer to this quantity as* **mixing**, *since it relates to mixed product monomials of the Taylor polynomial expansion. Since mixing is a property of the task, assessing the mixing ability of an* MPNN *becomes an additional metric of expressive power. Graph functions whose mixing violates* (7), **cannot** *be learned by* MPNNs *satisfying Theorem 3.2.*

## 4 Over-squashing limits the expressive power of MPNNs

Over-squashing was originally described in Alon & Yahav (2021) as the failure of MPNNs to propagate information across distant nodes. In fact, Topping et al. (2022); Black et al. (2023); Di Giovanni et al. (2023) showed that over-squashing – quantified by the sensitivity of node $v$ to the input feature at node $u$ via their Jacobian – is affected by topological properties such as curvature and effective resistance; however, no connection to how over-squashing affects the expressive power of MPNNs, nor proposal of measures for over-squashing have been derived. In light of these works, it is evident that over-squashing is related to the inability of MPNNs to model interactions among certain nodes, depending on the underlying graph topology. Since one can rely on the Taylor expansion of a graph function to measure such interactions through the second-order derivatives, i.e. the maximal mixing, we leverage Definition 3.1 to propose a **novel**, broader, but more accurate, characterization of over-squashing:

**Definition 4.1.** *Given the prediction $y_{\mathsf{G}}^{(m)}$ of an* MPNN *with capacity $(m, \mathsf{w})$, we define the* **pairwise** *over-squashing of $v, u$ as*

$$\mathsf{OSQ}_{v,u}(m, \mathsf{w}) = \left( \mathsf{mix}_{y_{\mathsf{G}}^{(m)}}(v, u) \right)^{-1}.$$

Our notion of over-squashing is a *pairwise* measure over the graph that naturally depends on the graph-topology, as well as the capacity of the model. In particular, it captures how over-squashing pertains to the ability of the model to mix (induce interactions) between features associated with different nodes. If such maximal mixing is large, then there is no obstruction to exchanging information between the given nodes and hence the over-squashing measure would be small; conversely, the over-squashing is large precisely when the model struggles to mix features associated with nodes $v$ and $u$.

In general though, computing the *actual* mixing induced by an MPNN may be difficult. Accordingly, we can rely on Theorem 3.2 to derive a proxy for the over-squashing measure that will be used to obtain necessary conditions on the capacity of an MPNN to induce a required level of mixing:

**Definition 4.2.** *Given an* MPNN *with capacity* $(m, \mathsf{w})$*, we approximate* $\mathsf{OSQ}_{v,u}(m, \mathsf{w})$ *by*

$$\widetilde{\mathsf{OSQ}}_{v,u}(m, \mathsf{w}) := \Big( \sum_{k=0}^{m-1} \mathsf{w}^{2m-k-1} \Big( \mathsf{w}(\mathbf{S}^{m-k})^{\top} \mathrm{diag}(\mathbf{1}^{\top}\mathbf{S}^k)\mathbf{S}^{m-k} + c^{(2)}\mathbf{Q}_k \Big)_{vu} \Big)^{-1} \leq \mathsf{OSQ}_{v,u}(m, \mathsf{w}).$$

The upper bound is a simple application of Theorem 3.2. To justify our characterization, we begin by considering simple settings. If the network has no bandwidth through the weights ($\mathsf{w} = 0$), then $\widetilde{\mathsf{OSQ}}_{v,u}(m, 0) = \infty$. Besides, the proposed measure captures the special case of *under-reaching* (Barceló et al., 2019), since $\widetilde{\mathsf{OSQ}}_{v,u}$ is infinite (i.e., zero mixing) whenever $2m < d_{\mathsf{G}}(v, u)$, where $d_{\mathsf{G}}$ is the shortest-walk distance on $\mathsf{G}$. We also recall that for simplicity we have taken $c_\sigma = 1$, but the measure extends to arbitrary non-linear activations $\sigma$. Finally, we generalize the characterization of $\mathsf{OSQ}$ to node-level tasks in Section E of the Appendix.

We can rephrase our novel approach for studying expressivity through pairwise mixing, in terms of the over-squashing measure and its proxy. By Theorem 3.2 we derive that a ***necessary condition for a smooth MPNN to learn a function*** $y_{\mathsf{G}}$ ***with mixing*** $\mathsf{mix}_{y_{\mathsf{G}}}(v, u)$ ***is***

$$\widetilde{\mathsf{OSQ}}_{v,u}(m, \mathsf{w}) < \left( \mathsf{mix}_{y_{\mathsf{G}}}(v, u) \right)^{-1}. \tag{8}$$

An MPNN of given capacity might suffer or not from over-squashing, *depending on the level of mixing required by the underlying task*. Over-squashing can then be understood as the **misalignment** between the task and the underlying topology, as measured by the gap between the maximal mixing induced by an MPNN over nodes $v, u$ and the mixing required by the task.

**Strategy.** For a given graph $\mathsf{G}$, to reduce the value of $\widetilde{\mathsf{OSQ}}$ and hence satisfy (8), the capacity $(m, \mathsf{w})$ must satisfy constraints posed by $\mathsf{G}$ and the choice of $v, u$. Since we can increase the capacity by taking either larger weights or more layers, we consider these two regimes separately. Below, we expand (8) in order to derive minimal requirements on the quantities $\mathsf{w}$ and $m$ to induce a certain level of mixing, thereby deriving lower bounds on the capacity of an MPNN to learn functions with given second-order derivatives (mixing).

**Choice of A.** For simplicity, we restrict our analysis to the case $\mathbf{A} = \mathbf{A}_{\mathrm{sym}} := \mathbf{D}^{-1/2}\mathbf{A}\mathbf{D}^{-1/2}$ and extend the results to $\mathbf{D}^{-1}\mathbf{A}$ and $\mathbf{A}$ in Section D of the Appendix. For the unnormalized adjacency, the bounds look more favourable, simply because we are no longer normalizing the messages, which makes the model more sensitive for all nodes. In Section D.4 we comment on how one could account for the lack of normalization by taking relative measurement for $\mathsf{OSQ}$ in the spirit of Xu et al. (2019).

## 4.1 The case of fixed depth $m$ and variable weights norm $\mathsf{w}$

To assess the ability of the norm of the weights $\mathsf{w}$ to increase the capacity of an MPNN and hence reduce $\widetilde{\mathsf{OSQ}}$, we consider the boundary case where the depth $m$ is the minimal required for an MPNN to have a non-zero mixing among $v, u$ (half the shortest-walk distance $d_{\mathsf{G}}$ between the nodes).

**Theorem 4.3.** *Let* $\mathbf{A} = \mathbf{A}_{\mathrm{sym}}$, $r := d_{\mathsf{G}}(v, u)$, $m = \lceil r/2 \rceil$, *and* $q$ *be the number of paths of length* $r$ *between* $v$ *and* $u$*. For an* MPNN *satisfying Theorem 3.2 with capacity* $(m = \lceil r/2 \rceil, \mathsf{w})$*, we find* $\widetilde{\mathsf{OSQ}}_{v,u}(m, \mathsf{w}) \cdot (c_2\mathsf{w})^r(\mathbf{A}^r)_{vu} \geq$

1. *In particular, if the* MPNN *generates mixing* $\mathsf{mix}_{y_\mathsf{G}}(v, u)$, *then*

$$\mathsf{w} \geq \frac{d_{\min}}{c_2} \left( \frac{\mathsf{mix}_{y_\mathsf{G}}(v, u)}{q} \right)^{\frac{1}{r}}.$$

Theorem 4.3 highlights that if the depth is set as the minimum required for *any* non-zero mixing, then the norm of the weights $\mathsf{w}$ has to be large enough depending on the connectivity of $\mathsf{G}$ – recall that for models as GCN, we have $c_2 = 1$. However, increasing $\mathsf{w}$ is not optimal and may lead to poorer generalization capabilities (Bartlett et al., 2017; Garg et al., 2020). Besides, controlling the maximal operator norm from below is, in general, tricky. A more reasonable strategy might be then to increase the number of layers; we explore such approach next, after reviewing a few explicit examples.

**Some examples.** We illustrate the bounds in Theorem 4.3 and for simplicity, we set $c_2 = 1$. Consider a *tree* $\mathsf{T}_d$ of arity $d$, with $v$ the root and $u$ a leaf at distance $r$ and depth $m = r/2$; then $\widetilde{\mathsf{OSQ}}_{v,u}(m, \mathsf{w}) \geq \mathsf{w}^{-r}(d+1)^{r-1}$ and the operator norm required to generate mixing $y(v, u)$ is

$$\mathsf{w} \geq (d+1) \left( \frac{y(v, u)}{d+1} \right)^{\frac{1}{r}}.$$

We note that by taking $d = 1$ we recover the case of the path-graph (1D grid). Since the operator norm of the weights grows with the branching factor, we see that, in general, the capacity required by MPNNs to solve long-range tasks could be higher on graphs than on sequences (Alon & Yahav, 2021). We also consider the case of a 1-layer MPNN on a complete graph $\mathsf{K}_n$ with $v \neq u$. We find that $\widetilde{\mathsf{OSQ}}_{v,u}(m, \mathsf{w}) \geq (n-1)/\mathsf{w}$ and hence the operator norm required to generate mixing $y(v, u)$ is $\mathsf{w} \geq (n-1)y(v, u)$. We note how the measure of over-squashing also captures the problem of redundancy of messages (Chen et al., 2022). In fact, even if $v, u$ are at distance 1, the more nodes are there in the complete graph and hence the more messages are exchanged, the more difficult for a shallow MPNN to induce enough mixing among those *specific* nodes.

## 4.2 The case of fixed weights norm $\mathsf{w}$ and variable depth $m$

We now study the (desirable) setting where $\mathsf{w}$ is bounded, and derive the depth necessary to induce mixing of nodes $v, u$. Below, we let $0 = \lambda_0 < \lambda_1 \leq \ldots \leq \lambda_{n-1}$ be the eigenvalues of the normalized graph Laplacian $\mathbf{\Delta} = \mathbf{I} - \mathbf{A}_{\mathrm{sym}}$; we note that $\lambda_1$ is the spectral gap and $\lambda_{n-1} < 2$ if $\mathsf{G}$ is not bipartite (Chung & Graham, 1997). We also recall that $d_\mathsf{G}$ is the shortest-walk distance and $\tau$ is the **commute time** (defined in Section 2). Finally, if $d_{\max}$ and $d_{\min}$ denote the maximal and minimal degrees, respectively, we set $\gamma := \sqrt{d_{\max}/d_{\min}}$.

**Theorem 4.4.** *Consider an* MPNN *satisfying Theorem 3.2, with* $\max\{\mathsf{w}, \omega/\mathsf{w} + c_1\gamma + c_2\} \leq 1$, *and* $\mathbf{A} = \mathbf{A}_{\mathrm{sym}}$. *If* $\widetilde{\mathsf{OSQ}}_{v,u}(m, \mathsf{w}) \cdot (\mathsf{mix}_{y_\mathsf{G}}(v, u)) \leq 1$, *i.e. the* MPNN *generates mixing* $\mathsf{mix}_{y_\mathsf{G}}(v, u)$ *among the features associated with nodes $v, u$, then the number of layers $m$ satisfies*

$$m \geq \frac{\tau(v, u)}{4c_2} + \frac{|\mathsf{E}|}{\sqrt{d_v d_u}} \left( \frac{\mathsf{mix}_{y_\mathsf{G}}(v, u)}{\gamma\mu} - \frac{1}{c_2} \left( \frac{\gamma + |1 - c_2\lambda^*|^{r-1}}{\lambda_1} + 2\frac{c^{(2)}}{\mu} \right) \right), \tag{9}$$

*where* $r = d_\mathsf{G}(v, u)$, $\mu = 1 + 2c^{(2)}(1 + \gamma)$ *and* $|1 - c_2\lambda^*| = \max_{0 < \ell \leq n-1} |1 - c_2\lambda_\ell| < 1$.

Theorem 4.4 provides a *necessary condition* on the depth of an MPNN to induce enough mixing among nodes $v, u$. We see that the MPNN must be sufficiently deep if the task depends on interactions between nodes at high commute time $\tau$ and in fact, if the required mixing is large enough, then $m \gtrsim \tau(v, u)$. Note that the lower bound on the depth can translate into a *practical impossibility statement*, since the commute time $\tau$ can be as large as $\mathcal{O}(n^3)$ (Chandra et al., 1996). We also recall that for MPNN models such as GCN, the bounds above simplify since $c^{(2)} = 0$ and hence $\mu = 1$. If we let $\mathrm{diam}(\mathsf{G})$ be the diameter of $\mathsf{G}$, Theorem 4.4 implies:

**Corollary 4.5.** *Given a graph with features, nodes $v, u$, and an* MPNN *satisfying Theorem 4.4, if the depth $m$ fails to satisfy* (9), *then the* MPNN *cannot learn functions with mixing* $\mathsf{mix}_{y_\mathsf{G}}(v, u)$.

Therefore, if (i) the graph is such that the commute time between $v, u$ is large, and (ii) the task depends on high-mixing of features associated with $v, u$, then **over-squashing limits the expressive power** of

MPNNs with bounded depth. In fact, increasing the depth based on the number of nodes may also cause issues connected to vanishing gradients (Di Giovanni et al., 2023) and over-smoothing (Rusch et al., 2023a; Cai & Wang, 2020; Rusch et al., 2023b; Di Giovanni et al., 2022). In contrast to existing analysis on the limitations of MPNNs via graph-isomorphism test, our results characterize the expressivity of MPNNs even when **meaningful (geometric) features are provided**. Corollary 4.5 implies that for any given maximal number of layers $m$ and for any graph $G$ with commute time $\tau$, we can identify classes of functions that are outside the hypothesis space of the MPNNs: these are the ones whose mixing violates (9).

Since the commute time of two adjacent nodes $v, u$ equals $2|\mathsf{E}|$ if $(v, u)$ is a **cut-edge** (Aleliunas et al., 1979), our result shows that MPNNs may require $m = \Omega(|\mathsf{E}|)$ to generate enough mixing along a cut-edge, drawing a connection with Zhang et al. (2023), where it was shown that most GNNs fail to identify cut-edges on *unattributed* graphs. Moreover, since the commute time is proportional to the effective resistance $\mathsf{R}$ by $2\mathsf{R}(v, u)|\mathsf{E}| = \tau(v, u)$, Theorem 4.4 connects recent works Di Giovanni et al. (2023); Black et al. (2023) relating the Jacobian of node features and $\mathsf{R}$, and the expressive power of MPNNs in terms of their mixing.

We emphasize how our theoretical results are far *more general* than assessing the inability of an MPNN to solve tasks with long-range interactions. Our analysis shows, precisely, how over-squashing can be understood as a fundamental problem associated with how hard – as measured by the number of layers required – is for an MPNN to exchange *information* between nodes that are 'badly connected', as per their commute time metric distance, whatever this *information* might be.

> **Message of the Section:** *If the mixing generated by an MPNN is small, then the model struggles to exchange information between nodes. Accordingly, we propose to measure over-squashing as the inverse of the mixing induced by an MPNN. Next, we asked: given a **target** level of mixing, what is the capacity of the MPNN required to match such quantity? We have addressed this question for the cases of bounded depth (Theorem 4.3) and bounded weights (Theorem 4.4). In the latter, we have shown that the minimal number of layers required by an MPNN to match a given level of mixing, grows with the commute time distance. Therefore, our analysis explicitly identifies classes of functions that are impossible to be learned on certain graphs for MPNNs of bounded depth.*

## 5 Experimental validation of the theoretical results

Next, we aim to empirically verify the theoretical findings of this paper, i.e., the characterization of the impact of the graph topology (via commute time $\tau$), the GNN architecture (depth, norm of weights), and the underlying task (node mixing) on over-squashing. This, however, requires detailed information about the underlying function to be learned, which is not readily available in practice. Hence, we perform our empirical test in a controlled environment, but at the same time, we base our experiments on the *real world* ZINC chemical dataset (Irwin et al., 2012) and follow the experimental setup in Dwivedi et al. (2020), constraining the number of molecular graphs to 12K. Moreover, we exclude the edge features from this experiment and fix the MPNN size to ∼100K parameters. However, instead of regressing the constrained solubility based on the molecular input graphs, we define our own synthetic node features as well as our own target values as follows.

Let $\{\mathsf{G}^i\}$ be the set of the 12K ZINC molecular graphs. We set all node features to zero, except for two, which are set to uniform random numbers $x_{u^i}^i, x_{v^i}^i$ between 0 and 1 (i.e., $x_{u^i}^i, x_{v^i}^i \sim \mathcal{U}(0, 1)$) for all $i$. The target output is set to $y^i = \tanh(x_{u^i}^i + x_{v^i}^i)$ for all $i$. Hence, the task entails a non-linear mixing with non-vanishing second derivatives. The two non-zero node features $x_{u^i}^i, x_{v^i}^i$ are positioned on $\mathsf{G}_i$ according to the commute time $\tau$, i.e., for a given $\alpha \in [0, 1]$, we choose the nodes $u^i, v^i$ as the $\alpha$-quantile of the $\tau$-distribution over $\mathsf{G}_i$. This enables us to have the desired control on the level of commute time of the underlying mixing (i.e., $\alpha_1 < \alpha_2$, if $\alpha_1$ results in a lower $\tau$ than $\alpha_2$). See Fig. 2 for an illustration of a ZINC graph together with the histogram of the corresponding $\tau$ between all pairs of different nodes. We call this graph dataset the *synthetic ZINC dataset.*

We consider four different MPNN models namely GCN (Kipf & Welling, 2017), GIN (Xu et al., 2019), GraphSAGE (Hamilton et al., 2017), and GatedGCN (Bresson & Laurent, 2017). Moreover, we choose the MAX-pooling as the GNN readout, which is supported by Theorem 3.2 and forces the GNNs to make use of the message-passing in order to learn the mixing.

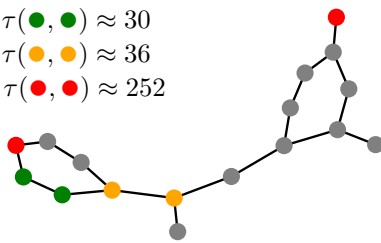
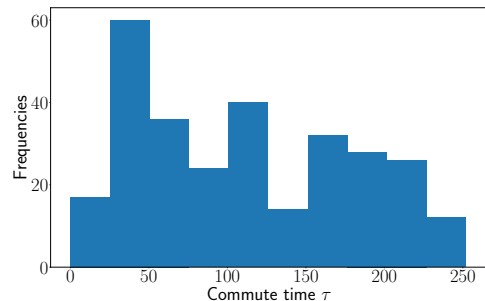

Figure 2: (**Left**) Exemplary molecular graph of the ZINC (12K) dataset with colored nodes corresponding to different values of commute time $\tau$. We note that $\tau$ is a more refined measure than the distance, and in fact beyond long-range nodes (red case), $\tau$ also captures other topological properties (yellow nodes are adjacent but belong to a *cut-edge, so their commute-time is* $2|\mathsf{E}|$). (**Right**) Histogram of commute time $\tau$ between all pairs of the graph nodes.

## 5.1 The role of commute time

In this task, we empirically analyse the effect of the commute time $\tau$ of the underlying mixing on the performance of the MPNNs. To this end, we fix the architecture for all considered MPNNs. In particular, we set the depth to $m = \max_i \lceil \mathrm{diam}(\mathsf{G}^i)/2 \rceil$, which happens to be $m = 11$ for the considered ZINC 12K graphs, such that the MPNNs are guaranteed *not to underreach*. We further vary the value of the $\alpha$-quantile of the $\tau$-distributions over the graphs $\mathsf{G}^i$ between 0 and 1, thus controlling the level of commute times. According to our theoretical findings in Section 4, the measure $\widetilde{\mathsf{OSQ}}_{v,u}$ (Definition 4.2) heavily depends on the commute time $\tau$ of the underlying mixing as derived in Theorem 4.4. In fact, this can be empirically verified in Appendix Fig. 7, where $\widetilde{\mathsf{OSQ}}_{v,u}$ increases for increasing values of $\alpha$. Thus, we would expect the MPNNs to perform significantly worse for increasing levels of the commute time. This is indeed confirmed in Fig. 3 depicting the resulting test mean absolute error (MAE) (average and standard deviation over several random weight initializations), and showing that the test MAE increases for larger values of $\alpha$ for all considered MPNNs. The same qualitative behavior can be observed for the *training* MAE (Appendix F.2). We further note that the models exhibit low standard deviation, as can be seen in Fig. 3 as well.

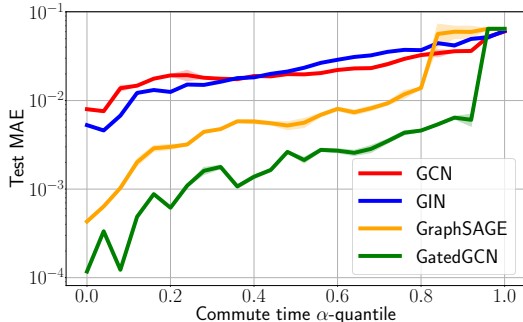
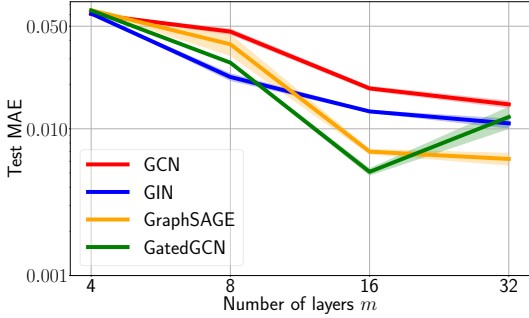
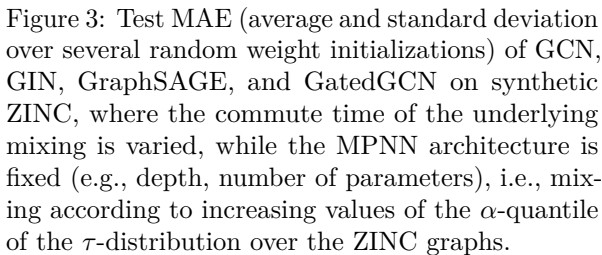

Figure 3: Test MAE (average and standard deviation over several random weight initializations) of GCN, GIN, GraphSAGE, and GatedGCN on synthetic ZINC, where the commute time of the underlying mixing is varied, while the MPNN architecture is fixed (e.g., depth, number of parameters), i.e., mixing according to increasing values of the $\alpha$-quantile of the $\tau$-distribution over the ZINC graphs.

Figure 4: Test MAE (average and standard deviation over several random weight initializations) of GCN, GIN, GraphSAGE, and GatedGCN on synthetic ZINC, where the commute time is fixed to be high (i.e., at the level of the 0.8-quantile), while only the depth of the underlying MPNN is varied between 4 and 32 (all other architectural components are fixed).

## 5.2  The role of depth

In this task, we study the effect of the depth on the performance of the MPNNs. To this end, we consider a high commute time-regime by setting $\alpha = 0.8$. Note that in this case the maximum (over all graphs $\mathsf{G}^i$) shortest path between two nodes $u^i, v^i$ is 14. Therefore, a *depth of $m = 7$ is sufficient to avoid under-reaching on all graphs*. However, according to the over-squashing measure we provide and the conclusions of Theorem 4.4, we expect the MPNNs to be able to induce more mixing among nodes $v, u$, and hence reduce the error, as we increase the number of layers. This expectation is further evidenced in Appendix Fig. 8, where the computed $\widetilde{\mathsf{OSQ}}$ decreases for increasing number of layers. In Fig. 4, we plot the test MAE (average and standard deviation over several random weight initializations) of all considered MPNNs for increasing number of layers. We can indeed see that all considered GNNs benefit from depth, and thus higher capacity (Definition 3.3), as GatedGCN obtains the lowest test MAE with 16 layers, as well as GraphSAGE, GIN, and GCN with 32 layers. The same qualitative behavior can be observed for the *training* MAE (Appendix F.2). Our theoretical results provide a strong explanation as to why a task **only** depending on the mixing of nodes within 14 hops – so that 7 layers would suffice – actually benefits from many more layers. Naturally, we cannot increase the depth arbitrarily, as at some point other issues emerge, such as vanishing gradients, which impact the trainability of the MPNNs (Rusch et al., 2022). Moreover, as can be additionally seen in Fig. 4, all models exhibit low standard deviation (over several trained MPNNs). We finally point out that, while over-squashing occurs – as can be seen when the number of layers 'only' equals the minimal one to avoid underreaching – it can be mitigated by taking much deeper architectures.

## 5.3  The role of mixing

We further test the considered MPNN architectures on their performance with respect to different mixings. To this end, we consider again the tanh-based mixing as in our previous tasks (i.e., regressing targets $y_i = \tanh(x^i_{u^i} + x^i_{v^i})$ for each graph $\mathsf{G}^i$ in the dataset), as well as another mixing based on the exponential function (i.e., with targets $y_i = \exp(x^i_{u^i} + x^i_{v^i})$). We note that these two tasks differ significantly in terms of their maximal mixing values (4) (shown in Table 1). Thus, according to (8) and Theorem 4.4, we would expect that MPNNs perform significantly worse in the case of higher maximal mixing, i.e., for the exponential-based mixing compared to the tanh-mixing. To confirm this empirically, we train the MPNNs on both types of mixing and provide the resulting relative MAEs (i.e., MAE divided by the $L^1$-norm of the targets) in Table 1. We can see that all four MPNNs perform significantly better on the tanh-mixing than on the exponential-based mixing. Moreover, increasing the range for the exponential-based mixing from 1 to 1.5 further impairs the performance of all considered MPNNs. In order to check if this difference in performance can simply be explained by a higher capacity required by a neural network to accurately approximate the mapping $\exp(x+y)$ compared to $\tanh(x+y)$ for some inputs $x, y \in \mathbb{R}$, we train a simple two-layer feed-forward neural network (with 2 inputs, i.e., $x$ and $y$) on both mappings. The trained networks reach a similarly low relative MAE of $4.6 \times 10^{-4}$ for the $\tanh(x + y)$ mapping as well as $4.1 \times 10^{-4}$ for the $\exp(x + y)$ mapping using an input range of $(0, 1)$ and $4.0 \times 10^{-4}$ for an input range of $(0, 1.5)$. Thus, we can conclude that the significant differences in the obtained results in Table 1 are not caused by a higher capacity required by a neural network to learn the underlying mappings of the different mixings.

Table 1: Relative MAE of GCN, GIN, GraphSAGE and GatedGCN on different choices of mixing on synthetic ZINC for a fixed 0.8-quantile of the commute time distributions over graphs $\mathsf{G}_i$.

| **Mixing** | *input interval* | *maximal mixing* | GCN | GIN | GraphSAGE | GatedGCN |
|---|---|---|---|---|---|---|
| $\tanh(x^i_{u^i} + x^i_{v^i})$ | $(0, 1)$ | $\approx 0.77$ | 0.024 | 0.014 | 0.006 | 0.004 |
| $\exp(x^i_{u^i} + x^i_{v^i})$ | $(0, 1)$ | $\approx 7.4$ | 0.043 | 0.021 | 0.033 | 0.008 |
| $\exp(x^i_{u^i} + x^i_{v^i})$ | $(0, 1.5)$ | $\approx 20.1$ | 0.054 | 0.035 | 0.075 | 0.014 |

# 6 Discussion

**Related Work: expressive power of MPNNs.** The MPNN class in (2) is as **powerful** as the 1-WL test (Weisfeiler & Leman, 1968) in distinguishing *unattributed* graphs (Xu et al., 2019; Morris et al., 2019). In fact, MPNNs typically struggle to compute graph properties on feature-less graphs (Dehmamy et al., 2019; Chen et al., 2020; Sato et al., 2019; Loukas, 2020). The expressivity of GNNs has also been studied from the point of view of logical and tensor languages (Barceló et al., 2019; Azizian & Lelarge, 2021; Geerts & Reutter, 2022). Nonetheless, far less is known about which functions of node features MPNNs can learn and *the capacity required to do so.* Razin et al. (2022) recently studied the separation rank of a specific MPNN class. While this approach is a strong inspiration for our work, the results in Razin et al. (2022) only apply to a single unconventional family of MPNNs which does not include MPNN models used in practice. *Our results instead hold in the full generality of* (2) *and provide a novel approach for investigating the expressivity of MPNNs through the mixing they are able to generate among features.*

**Mixing and the WL test.** We emphasize that throughout our analysis we had **no** assumption on the nature of the features, that can in fact be structural or positional – meaning that the MPNNs we have considered above, may also be more powerful than the 1-WL test. Our derivations do not rely on the ability to distinguish different node representations, but rather on the ability of the MPNN to mix information associated with different nodes. This novel alternative paradigm may help design GNNs that are more powerful at mixing than MPNNs, and may further shed light on how and when frameworks such as Graph Transformers can solve the underlying task better than conventional MPNNs. Similarly, our analysis holds for all equivariant MPNNs of the form (2) acting on geometric graphs embedded in Euclidean space, independent of the geometric information included in the features.

**Differences between our results and existing works on over-squashing.** The problem of **over-squashing** was introduced in Alon & Yahav (2021) and studied through sensitivity analysis in Topping et al. (2022). This approach was generalized in Black et al. (2023); Di Giovanni et al. (2023) who proved that the Jacobian of node features is likely to be small if the nodes have high commute time (effective resistance). We discuss more in detail the novelty of this work when compared to Topping et al. (2022); Di Giovanni et al. (2023). (i) In Topping et al. (2022); Di Giovanni et al. (2023) there is no analysis on which functions MPNNs cannot learn as a consequence of over-squashing, nor a formal measure of over-squashing is provided. In fact, they only derive some conditions under which the Jacobian of node features can be small, but do not connect it to the ability of MPNNs to learn desired function classes. Besides, the Jacobian of node features may not be suited for studying over-squashing for graph-level tasks. Note that our theory holds for node-level tasks and generalize existing approaches (see Appendix E). (ii) The analysis in Topping et al. (2022) does not study over-squashing among nodes at distance larger than 2 and does not provide insights on the capacity required to learn certain tasks. (iii) Finally, the analysis in Di Giovanni et al. (2023) does not account for MPNNs such as GatedGCN, and is only carried out in a simplified setting where the activation function is chosen to be a ReLU and its derivatives are taken to be the same, in expectation, for every path in the computational graph. *We have extended these ideas to connect over-squashing and expressive power by studying higher-order derivatives of the MPNN and relating them to the capacity of the model and the underlying graph-topology.*

## 6.1 Limitations and ways forward

**The measures OSQ and $\widetilde{\mathsf{OSQ}}$.** Definition 4.1 considers pairs of nodes and second-order derivatives; this could be generalized to build a hierarchy of measures with increasingly higher-order interactions of nodes. The proxy (lower bound) $\widetilde{\mathsf{OSQ}}$ in Definition 4.2 allows to derive necessary conditions for the MPNN to learn classes of functions based on their mixing. If, depending on the problem, one has access to better estimates on the mixing induced by an MPNN than (7), then one can extend our approach and get a finer approximation of OSQ. In the paper we focused our analysis on the case of graph-level tasks; we provide an extension to node-level functions in Section E.

**Beyond sum-aggregations.** Our results apply to MPNNs as in (2), where **A** is constant, and do not include attention-based MPNNs (Veličković et al., 2018; Brody et al., 2022) or Graph-Transformers (Kreuzer et al., 2021; Mialon et al., 2021; Ying et al., 2021; Rampasek et al., 2022) which further depend on features via

normalization. Extending the analysis to these models is only more technically involved. More generally, one could replace the aggregation $\sum_u \mathsf{A}_{vu}$ in (2) with a smooth, permutation invariant operator $\bigoplus$ (Bronstein et al., 2021; Ong & Veličković, 2022). *Our formalism will then prove useful to assess if different aggregations are more expressive in terms of the mixing (interactions) they are able to generate.*

**Graph rewiring.** Another way of going beyond (2) to find MPNNs with lower $\mathsf{OSQ}$ is to replace **A** with a different matrix **A**′, (partly) independent of the connectivity of the input graph. In other words, one could rewire the graph, by replacing $\mathsf{G}$ with $\mathsf{G}'$. Indeed, Theorem 4.4 further validates why recent graph-rewiring methods such as (Arnaiz-Rodríguez et al., 2022; Karhadkar et al., 2023; Deac et al., 2022; Black et al., 2023) manage to alleviate over-squashing: by adding edges that decrease the overall effective resistance (commute time) of the graph, these methods reduce the measure $\mathsf{OSQ}$. More generally, *Definition 4.2 allows one to measure whether a given rewiring is beneficial in terms of over-squashing (and hence of the mixing generated) and to what extent.* In fact, it follows from Theorem 4.4 that methods like (Deac et al., 2022; Shirzad et al., 2023) are **optimal** from the point of view of friendliness of the computational graph to message-passing, since they both propose to propagate information over *expander* graphs, which are sparse and have commute time scaling linearly with the number of edges. Finally, our theoretical results suggest that for data given by point clouds in 3D space, the choice of a computational graph over which message passing can operate, should also account for the its commute time, given that the latter is closely aligned with over-squashing.

**Acknowledgements.** The authors would like to thank Dr. Martín Arjovsky, Dr. Charles Blundell and Dr. Karl Tuyls for their insightful feedback and constructive suggestions on an earlier version of the manuscript.

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

## A  Outline of the appendix

We provide an overview of the appendix. Since in the appendix we report additional theoretical results and considerations, we first point out to the most relevant content: the proofs of the main results, the extension of our discussion and analysis to node-level tasks, and the additional ablation studies.

**Where to find proofs of the main results.** We prove Theorem 3.2 in Section C.1, we prove Theorem 4.3 in Section D.1, and finally we prove Theorem 4.4 in Section D.3.

**Where to find the extension to node-level tasks.**Concerning the case of node-level tasks, we present a thorough discussion on the matter in Section E, where we extend the definition of the over-squashing measure and generalize Theorem 3.2 and Theorem 4.4 to node-level predictions of the MPNN class in (2).

**Where to find additional ablation studies.** In Section F we have conducted further experiments on the profile of the over-squashing measure $\widetilde{\mathsf{OSQ}}$ across different MPNN models as well as on the training mean average error, to further validate our claims on over-squashing hindering the expressive power of MPNNs.

Next, we summarize the contents of the Appendix more in detail below.

- In order to be self-consistent, in Section B we review important notions pertaining to the spectrum of the graph-Laplacian and known properties of random walks on graphs, that will be then be used in our proofs.

- In Section C we prove the main theorem on the maximal mixing induced by MPNNs (Theorem 3.2). In particular, we also derive additional results on the mixing generated at a specific node, which will turn out useful when extending the characterization of the over-squashing measure $\widetilde{\mathsf{OSQ}}$ for node-level tasks.

- In Section D we prove the main results of Section 4, mainly Theorem 4.3 and Theorem 4.4. Further, we also derive an explicit (sharper) characterization of the depth required to induce enough mixing among nodes, in terms of the pseudo-inverse of the graph-Laplacian. Finally, in Section D.4 we extend the results to the case of the unnormalized adjacency matrix and discuss relative over-squashing measures.

- In Section E we generalize the over-squashing measure for node-level tasks, commenting on the differences between our approach and existing works (mainly (Topping et al., 2022; Black et al., 2023; Di Giovanni et al., 2023)). In particular, we show that the same conclusions of Theorem 4.4 hold for node-level predictions too.

- Finally, in Section F we report additional details on our experimental setup and further ablation studies concerning the over-squashing measure $\widetilde{\mathsf{OSQ}}$.

## B  Summary of spectral properties on graphs

**Basic notions of spectral theory on graphs.**

Throughout the appendix, we let $\boldsymbol{\Delta}$ be the normalized graph Laplacian defined by $\boldsymbol{\Delta} = \mathbf{I} - \mathbf{D}^{-1/2}\mathbf{A}\mathbf{D}^{-1/2}$. It is known Chung & Graham (1997) that the graph Laplacian is a symmetrically, positive semi-definite matrix whose spectral decomposition takes the form

$$\boldsymbol{\Delta} = \sum_{\ell=0}^{n-1} \lambda_\ell \phi_\ell \phi_\ell^\top, \tag{10}$$

where $\{\phi_\ell\}$ is an orthonormal basis in $\mathbb{R}^n$ and $0 = \lambda_0 < \lambda_1 < \ldots < \lambda_{n-1}$ – recall that since we assume $\mathsf{G}$ to be connected, the zero eigenvalue has multiplicity one, i.e. $\lambda_1 > 0$. We also note that we typically write $\phi_\ell(v)$ for the value of $\phi_\ell$ at $v \in \mathsf{V}$, and that the kernel of $\boldsymbol{\Delta}$ is spanned by $\phi_0$ with $\phi_0(v) = \sqrt{d_v/2|\mathsf{E}|}$. the results would extend to the bipartite cas As usual when doing spectral analysis one too if graphs, we exclude

the edge case of the bipartite graph to make sure that the largest eigenvalue of the graph Laplacian satisfies $\lambda_{n-1} < 2$ – yet all results hold for the bipartite case too provided we take $\|\nabla_2\psi\| < 1$. Finally, we let $\boldsymbol{\Delta}^\dagger$ denote the pseudo-inverse of the graph Laplacian, which can be written as

$$\boldsymbol{\Delta}^\dagger = \sum_{\ell=1}^{n-1} \frac{1}{\lambda_\ell} \phi_\ell \phi_\ell^\top, \tag{11}$$

and we emphasize that the sum starts from $\ell = 1$ since we need to ignore the kernel of $\boldsymbol{\Delta}$ spanned by the orthonormal vector $\phi_0$.

**Basic properties of Random Walks on graphs.** A simple Random Walk (RW) on $\mathsf{G}$ is a Markov chain supported on the nodes $\mathsf{V}$ with transition matrix defined by $\mathbf{P}(v, u) = d_v^{-1}$. While a RW can be studied through different properties, the one we are interested in is the *commute time $\tau$*, which represents the expected number of steps for a RW starting at $v$, to visit $u$ and then come back to $v$. The commute time is a *distance* on the graph and captures the diffusion properties associated with the underlying topology. In fact, while nodes that are distant often have larger commute time, the latter is more expressive than the shortest-walk graph-distance, since it also accounts (for example) for the number of paths connecting two given nodes. Thanks to Lovász (1993), we can write down the commute time among two nodes using the spectral representation of the graph Laplacian in (10):

$$\tau(v, u) = 2|\mathsf{E}| \sum_{\ell=1}^{n-1} \frac{1}{\lambda_\ell} \left( \frac{\phi_\ell(v)}{\sqrt{d_v}} - \frac{\phi_\ell(u)}{\sqrt{d_u}} \right)^2. \tag{12}$$

## C Proofs and additional details of Section 3

The goal of this section amounts to proving Theorem 3.2. To work towards this result, we first derive bounds on the Jacobian and Hessian of a single node feature after $m$ layers before the readout READ operation. We emphasize that our analysis below is novel, when compared to previous works of Black et al. (2023); Di Giovanni et al. (2023), on many accounts. First, Black et al. (2023); Di Giovanni et al. (2023) do not consider higher (second) order derivatives, limiting their discussion to the case of first order derivatives, which are not suited to capture notions of mixing among features – we will expand on this topic in Section E. Second, even for the case of first-order derivatives, our result below is more general since it holds for all MPNNs as in (2), which includes (i) message-functions $\psi$ that also depend on the input features (as for GatedGCN), and (ii) choices of message-passing matrices $\mathbf{A}$ that could be weighted and (or) asymmetric. Third, the analysis in Black et al. (2023); Di Giovanni et al. (2023) does not account for the role of the readout map and hence fails to study the expressive power of graph-level prediction of MPNNs as measured by the mixing they generate among nodes.

**Conventions and notations for the proofs.** First, we recall that $\mathbf{h}_v^{(0)} = \mathbf{x}_v \in \mathbb{R}^d$ is the input feature at node $v$. Below, we write $h_v^{(t),\alpha}$ for the $\alpha$-th entry of the feature $\mathbf{h}_v^{(t)}$. To simplify the notations, we rewrite the layer-update in (2) using coordinates as

$$h_v^{(t),\alpha} = \sigma(\tilde{h}_v^{(t-1),\alpha}), \quad 1 \le \alpha \le d, \tag{13}$$

where $\tilde{h}_v^{(t-1),\alpha}$ is the entry $\alpha$ of the pre-activated feature of node $v$ at layer $t$. We also let $\partial_{1,p}\psi^{(t),r}$ and $\partial_{2,p}\psi^{(t),r}$ be the $p$-th derivative of $(\psi^{(t)}(\cdot, x))_r$ and of $(\psi^{(t)}(x, \cdot))_r$, respectively. To avoid cumbersome notations, we usually omit to write the arguments of the derivatives of the message-functions $\psi$. Similarly, we let $\nabla_1\psi$ ($\nabla_2\psi$) be the $d \times d$ Jacobian matrix of $\psi$ with respect to the first (second) variable. Finally, given nodes $i, v, u \in \mathsf{V}$ we introduce the following terms:

$$\nabla_u \mathbf{h}_v^{(m)} := \frac{\partial \mathbf{h}_v^{(m)}}{\partial \mathbf{x}_u} \in \mathbb{R}^{d \times d}, \quad \nabla_{uv}^2 \mathbf{h}_i^{(m)} := \frac{\partial^2 \mathbf{h}_i^{(m)}}{\partial \mathbf{x}_u \partial \mathbf{x}_v} \in \mathbb{R}^{d \times (d \times d)}.$$

First, we derive an upper bound on the first-order derivatives of the node-features. This will provide useful to derive the more general second-order estimate of the MPNN-prediction. We highlight that the result

below extends the analysis in (Di Giovanni et al., 2023) to MPNNs with arbitrary (i.e. non-linear) message functions $\psi$, such as GatedGCN (Bresson & Laurent, 2017).

**Theorem C.1.** *Given MPNNs as in* (2), *let* $\sigma$ *and* $\psi^{(t)}$ *be* $\mathcal{C}^1$ *functions and assume* $|\sigma'| \leq c_\sigma$, $\|\Omega^{(t)}\| \leq \omega$, $\|\mathbf{W}^{(t)}\| \leq \mathsf{w}$, $\|\nabla_1\psi^{(t)}\| \leq c_1$, *and* $\|\nabla_2\psi^{(t)}\| \leq c_2$. *Let* $\mathbf{S} \in \mathbb{R}^{n \times n}$ *be*

$$\mathbf{S} := \frac{\omega}{\mathsf{w}}\mathbf{I} + c_1\mathrm{diag}(\mathbf{A1}) + c_2\mathbf{A}.$$

*Given nodes* $v, u \in \mathsf{V}$ *and* $m$ *the number of layers, the following holds:*

$$\|\nabla_u\mathbf{h}_v^{(m)}\| \leq (c_\sigma\mathsf{w})^m(\mathbf{S}^m)_{vu}. \tag{14}$$

*Proof.* Recall that the dimension of the features is taken to be $d$ for any layer $1 \leq t \leq m$. We proceed by induction. If $m = 1$ and we fix entries $1 \leq \alpha, \beta \leq d$, then using the shorthand in (13), we obtain

$$(\nabla_u\mathbf{h}_v^{(1)})_{\alpha\beta} = \sigma'(\tilde{h}_v^{(0),\alpha})\Big(\Omega_{\alpha\beta}^{(1)}\delta_{vu} + \sum_r W_{\alpha r}^{(1)}\sum_j \mathsf{A}_{vj}\Big(\partial_{1,\beta}\psi^{(1),r}\delta_{vu} + \partial_{2,\beta}\psi^{(1),r}\delta_{ju}\Big)\Big)$$

$$= \Big(\mathrm{diag}(\sigma'(\tilde{\mathbf{h}}_v^{(0)}))\Big(\mathbf{\Omega}^{(1)}\delta_{vu} + \mathbf{W}^{(1)}\Big(\sum_j \mathsf{A}_{vj}\delta_{vu}\nabla_1\psi^{(1)} + \mathsf{A}_{vu}\nabla_2\psi^{(1)}\Big)\Big)\Big)_{\alpha\beta}.$$

Therefore, we can bound the (spectral) norm of the Jacobian on the left hand side by

$$\|\nabla_u\mathbf{h}_v^{(1)}\| \leq \|\mathrm{diag}(\sigma'(\tilde{\mathbf{h}}_v^{(0)}))\|\Big(\|\mathbf{\Omega}^{(1)}\|\delta_{vu} + \|\mathbf{W}^{(1)}\|(c_1\sum_j \mathsf{A}_{vj}\delta_{vu} + c_2\mathsf{A}_{vu})\Big)$$

$$\leq c_\sigma(\omega\delta_{vu} + \mathsf{w}(c_1\sum_j \mathsf{A}_{vj}\delta_{vu} + c_2\mathsf{A}_{vu})) = c_\sigma\mathsf{w}\mathsf{S}_{vu},$$

which proves the estimate on the Jacobian for the case of $m = 1$. We now take the induction step, and follow the same argument above to write the node Jacobian after $m$ layers as

$$(\nabla_u\mathbf{h}_v^{(m)})_{\alpha\beta} = \sigma'(\tilde{h}_v^{(m-1),\alpha})\Big(\sum_r \Omega_{\alpha r}^{(m)}(\nabla_u\mathbf{h}_v^{(m-1)})_{r\beta}\Big)$$

$$+ \sigma'(\tilde{h}_v^{(m-1),\alpha})\Big(W_{\alpha r}^{(m)}\sum_j \mathsf{A}_{vj}\sum_p \Big(\partial_{1,p}\psi^{(m),r}(\nabla_u\mathbf{h}_v^{(m-1)})_{p\beta} + \partial_{2,p}\psi^{(m),r}(\nabla_u\mathbf{h}_j^{(m-1)})_{p\beta}\Big)\Big)$$

$$= \Big(\mathrm{diag}(\sigma'(\tilde{\mathbf{h}}_v^{(m-1)}))\mathbf{\Omega}^{(m)}\nabla_u\mathbf{h}_v^{(m-1)}\Big)_{\alpha\beta}$$

$$+ \Big(\mathrm{diag}(\sigma'(\tilde{\mathbf{h}}_v^{(m-1)}))\mathbf{W}^{(m)}\Big(\sum_j \mathsf{A}_{vj}\nabla_1\psi^{(m)}\nabla_u\mathbf{h}_v^{(m-1)} + \mathsf{A}_{vj}\nabla_2\psi^{(m)}\nabla_u\mathbf{h}_j^{(m-1)}\Big)\Big)_{\alpha\beta}.$$

Therefore, we can use the induction step to bound the Jacobian as

$$\|\nabla_u\mathbf{h}_v^{(m)}\| \leq c_\sigma\omega(c_\sigma\mathsf{w})^{m-1}(\mathbf{S}^{m-1})_{vu} + (c_\sigma\mathsf{w})^m\Big(\sum_j \mathsf{A}_{vj}c_1(\mathbf{S}^{m-1})_{vu} + \mathsf{A}_{vj}c_2(\mathbf{S}^{m-1})_{ju}\Big)$$

$$= (c_\sigma\mathsf{w})^m\Big(\Big(\frac{\omega}{\mathsf{w}}\mathbf{I} + c_1\mathrm{diag}(\mathbf{A1}) + c_2\mathbf{A}\Big)(\mathbf{S}^{m-1})\Big)_{vu} = (c_\sigma\mathsf{w})^m(\mathbf{S}^m)_{vu},$$

which completes the proof for the first-order bounds. $\square$

Before we move to the second-order estimates, we introduce some additional preliminary notations. Given nodes $i, v, u$, a matrix $\mathbf{S} \in \mathbb{R}^{n \times n}$ – which will always be chosen as per (5) – and an integer $\ell$, we write

$$\mathsf{P}_{i(vu)}^{(\ell)} := (\mathbf{S}^\ell)_{iv}(\mathbf{AS}^\ell)_{iu} + (\mathbf{S}^\ell)_{iu}(\mathbf{AS}^\ell)_{iv} + \sum_j (\mathbf{S}^\ell)_{jv}\,(\mathrm{diag}(\mathbf{A1}) + \mathbf{A})_{ij}\,(\mathbf{S}^\ell)_{ju}. \tag{15}$$

In particular, we denote by $\mathsf{P}_{(vu)}^{(\ell)} \in \mathbb{R}^n$ the vector with entries $(\mathsf{P}_{(vu)}^{(\ell)})_i = \mathsf{P}_{i(vu)}^{(\ell)}$, for $1 \leq i \leq n$.

**Theorem C.2.** *Given MPNNs as in* (2), *let* $\sigma$ *and* $\psi^{(t)}$ *be* $\mathcal{C}^2$ *functions and assume* $|\sigma'|, |\sigma''| \leq c_\sigma$, $\|\mathbf{\Omega}^{(t)}\| \leq \omega$, $\|\mathbf{W}^{(t)}\| \leq \mathsf{w}$, $\|\nabla_1\psi^{(t)}\| \leq c_1$, $\|\nabla_2\psi^{(t)}\| \leq c_2$, $\|\nabla^2\psi^{(t)}\| \leq c^{(2)}$. *Let* $\mathsf{S} \in \mathbb{R}^{n \times n}$ *be*

$$\mathsf{S} := \frac{\omega}{\mathsf{w}}\mathbf{I} + c_1\mathrm{diag}(\mathbf{A1}) + c_2\mathbf{A}.$$

*Given nodes* $i, v, u \in \mathsf{V}$, *if* $\mathsf{P}^{(\ell)}_{(vu)} \in \mathbb{R}^n$ *is as in* (15) *and* $m$ *is the number of layers, then we derive*

$$\|\nabla^2_{uv}\mathbf{h}^{(m)}_i\| \leq \sum_{k=0}^{m-1}\sum_{j \in \mathsf{V}}(c_\sigma\mathsf{w})^{2m-k-1}\,\mathsf{w}(\mathsf{S}^{m-k})_{jv}(\mathsf{S}^k)_{ij}(\mathsf{S}^{m-k})_{ju}$$

$$+ c^{(2)}\sum_{\ell=0}^{m-1}(c_\sigma\mathsf{w})^{m+\ell}(\mathsf{S}^{m-1-\ell}\mathsf{P}^{(\ell)}_{(vu)})_i. \tag{16}$$

*Proof.* First, we note that $\nabla^2_{uv}\mathbf{h}^{(m)}_i$ is a matrix of dimension $\mathbb{R}^{d \times (d \times d)}$. We then use the following ordering for indexing the columns – which is consistent with a typical way of labelling columns of the Kronecker product of matrices, as detailed below (note that indices here start from 1):

$$\frac{\partial^2 h^{(m),\alpha}_i}{\partial x^\beta_v \partial x^\gamma_u} := \left(\nabla^2_{uv}\mathbf{h}^{(m)}_i\right)_{\alpha,d(\beta-1)+\gamma}. \tag{17}$$

As above, we proceed by induction and start from the case $m = 1$:

$$\left(\nabla^2_{uv}\mathbf{h}^{(1)}_i\right)_{\alpha,d(\beta-1)+\gamma} = \sigma''(\tilde{h}^{(0),\alpha}_i)\Big(\Omega^{(1)}_{\alpha\gamma}\delta_{iv} + \sum_r W^{(1)}_{\alpha r}\sum_j \mathsf{A}_{ij}(\delta_{iv}\partial_{1,\gamma}\psi^{(1),r} + \delta_{jv}\partial_{2,\gamma}\psi^{(1),r})\Big)$$

$$\times \Big(\Omega^{(1)}_{\alpha\beta}\delta_{iu} + \sum_r W^{(1)}_{\alpha r}\sum_j \mathsf{A}_{ij}(\delta_{iu}\partial_{1,\beta}\psi^{(1),r} + \delta_{ju}\partial_{2,\beta}\psi^{(1),r})\Big)$$

$$+ \sigma'(\tilde{h}^{(0),\alpha}_i)\sum_r W^{(1)}_{\alpha r}\Big(\sum_j \mathsf{A}_{ij}\delta_{iu}(\partial_{1,\gamma}\partial_{1,\beta}\psi^{(1),r}\delta_{iv} + \partial_{2,\gamma}\partial_{1,\beta}\psi^{(1),r}\delta_{jv})\Big)$$

$$+ \sigma'(\tilde{h}^{(0),\alpha}_i)\sum_r W^{(1)}_{\alpha r}\Big(\mathsf{A}_{iu}(\partial_{1,\gamma}\partial_{2,\beta}\psi^{(1),r}\delta_{iv} + \partial_{2,\gamma}\partial_{2,\beta}\psi^{(1),r}\delta_{uv})\Big)$$

$$:= (Q_1)_{\alpha,\beta,\gamma} + (Q_2)_{\alpha,\beta,\gamma} + (Q_3)_{\alpha,\beta,\gamma},$$

where $Q_1$ is the term containing second derivatives of $\psi$ while $Q_2, Q_3$ are the remaining expressions including second-order derivatives of the message functions $\psi$. Using the same strategy as for the first-order estimates, we can rewrite the first term $Q_1$ as

$$(Q_1)_{\alpha,\beta,\gamma} = \Big(\mathrm{diag}(\sigma''(\tilde{\mathbf{h}}^{(0)}_v))\Big(\mathbf{\Omega}^{(1)}\delta_{iv} + \mathbf{W}^{(1)}(\sum_j \mathsf{A}_{ij}\delta_{iv}\nabla_1\psi^{(1)} + \mathsf{A}_{iv}\nabla_2\psi^{(1)})\Big)\Big)_{\alpha\gamma}$$

$$\times \Big(\mathbf{\Omega}^{(1)}\delta_{iu} + \mathbf{W}^{(1)}(\sum_j \mathsf{A}_{ij}\delta_{iu}\nabla_1\psi^{(1)} + \mathsf{A}_{iu}\nabla_2\psi^{(1)})\Big)_{\alpha\beta}$$

We now observe that given two matrices $\mathbf{B}, \mathbf{C} \in \mathbb{R}^{d \times d}$ and $1 \leq \alpha, \alpha', \beta, \gamma \leq d$, the entries of the Kronecker product $\mathbf{B} \otimes \mathbf{C}$ can be indexed as

$$(\mathbf{B} \otimes \mathbf{C})_{d(\alpha-1)+\alpha',d(\beta-1)+\gamma} = B_{\alpha\beta}C_{\alpha'\gamma}.$$

We now introduce the $d \times (d \times d)$ sub-matrix of $\mathbf{B} \otimes \mathbf{C}$ defined by

$$(\mathbf{B} \otimes \mathbf{C})'_{\alpha,d(\beta-1)+\gamma} = B_{\alpha\beta}C_{\alpha\gamma}. \tag{18}$$

Therefore, we can rewrite $(Q_1)_{\alpha,\beta,\gamma}$ as the entry $(\alpha, d(\beta - 1) + \gamma)$ of the $d \times (d \times d)$ sub-matrix

$$(\mathbf{Q}_1)_{\alpha,d(\beta-1)+\gamma} = (\mathbf{B} \otimes \mathbf{C})'_{\alpha,d(\beta-1)+\gamma}, \tag{19}$$

where

$$\mathbf{B} := \mathrm{diag}(\sigma''(\tilde{\mathbf{h}}_v^{(0)}))\Big(\mathbf{\Omega}^{(1)}\delta_{iv} + \mathbf{W}^{(1)}(\sum_j \mathsf{A}_{ij}\delta_{iv}\nabla_1\psi^{(1)} + \mathsf{A}_{iv}\nabla_2\psi^{(1)})\Big),$$

$$\mathbf{C} := \mathbf{\Omega}^{(1)}\delta_{iu} + \mathbf{W}^{(1)}(\sum_j \mathsf{A}_{ij}\delta_{iu}\nabla_1\psi^{(1)} + \mathsf{A}_{iu}\nabla_2\psi^{(1)}).$$

Next, we proceed to write $(Q_2)_{\alpha,\beta,\gamma}$ in matricial form. Before we do that, we observe that the Hessian of the message functions $(\mathbf{x}_i, \mathbf{x}_j) \mapsto \psi^{(t)}(\mathbf{x}_i, \mathbf{x}_j)$ takes the form

$$\nabla^2\psi^{(t)} = \begin{pmatrix} \nabla_{11}^2\psi^{(t)} & \nabla_{12}^2\psi^{(t)} \\ \nabla_{21}^2\psi^{(t)} & \nabla_{22}^2\psi^{(t)} \end{pmatrix},$$

where $\nabla_{ab}^2\psi^{(t)} \in \mathbb{R}^{d\times(d\times d)}$ and is indexed as follows

$$(\nabla_{ab}^2\psi^{(t)})_{r,d(\beta-1)+\gamma} = \partial_{a,\beta}\partial_{b,\gamma}\psi^{(t),r},$$

where $a, b \in \{1, 2\}$. Using these notations, we note that

$$\sum_r W_{\alpha r}^{(1)}\partial_{1,\gamma}\partial_{1,\beta}\psi^{(1),r} = \left(\mathbf{W}^{(1)}\nabla_{11}^2\psi^{(1)}\right)_{\alpha,d(\beta-1)+\gamma}.$$

Therefore, we derive

$$(Q_2)_{\alpha,\beta,\gamma} = (\mathbf{Q}_2)_{\alpha,d(\beta-1)+\gamma} = \sum_j \mathsf{A}_{ij}\delta_{iu}\delta_{iv}(\mathrm{diag}(\sigma'(\tilde{\mathbf{h}}_i^{(0)}))\mathbf{W}^{(1)}\nabla_{11}^2\psi^{(1)})_{\alpha,d(\beta-1)+\gamma}$$
$$+ \mathsf{A}_{iv}\delta_{iu}(\mathrm{diag}(\sigma'(\tilde{\mathbf{h}}_i^{(0)}))\mathbf{W}^{(1)}\nabla_{12}^2\psi^{(1)})_{\alpha,d(\beta-1)+\gamma}. \tag{20}$$

A similar argument works for $Q_3$:

$$(Q_3)_{\alpha,\beta,\gamma} = (\mathbf{Q}_3)_{\alpha,d(\beta-1)+\gamma} = \mathsf{A}_{iu}\delta_{iv}(\mathrm{diag}(\sigma'(\tilde{\mathbf{h}}_i^{(0)}))\mathbf{W}^{(1)}\nabla_{21}^2\psi^{(1)})_{\alpha,d(\beta-1)+\gamma}$$
$$+ \mathsf{A}_{iu}\delta_{uv}(\mathrm{diag}(\sigma'(\tilde{\mathbf{h}}_i^{(0)}))\mathbf{W}^{(1)}\nabla_{22}^2\psi^{(1)})_{\alpha,d(\beta-1)+\gamma}. \tag{21}$$

Therefore, we can combine (19), (20), and (21) to write

$$\|\nabla_{uv}^2\mathbf{h}_i^{(1)}\| \le \|\mathbf{Q}_1\| + \|\mathbf{Q}_2\| + \|\mathbf{Q}_3\|$$
$$\le c_\sigma\left(\omega\delta_{iv} + \mathsf{w}(c_1\mathrm{diag}(\mathbf{A}\mathbf{1})_i\delta_{iv} + c_2\mathsf{A}_{iv})\right)\left(\omega\delta_{iu} + \mathsf{w}(c_1\mathrm{diag}(\mathbf{A}\mathbf{1})_i\delta_{iu} + c_2\mathsf{A}_{iu})\right)$$
$$+ c_\sigma\mathsf{w}c^{(2)}\left(\mathrm{diag}(\mathbf{A}\mathbf{1})_i\delta_{iv}\delta_{iu} + \mathsf{A}_{iv}\delta_{iu}\right)$$
$$+ c_\sigma\mathsf{w}c^{(2)}\left(\mathsf{A}_{iu}\delta_{iv} + \mathsf{A}_{iu}\delta_{uv}\right).$$

Finally, we can rely on (15) to re-arrange the equation above as

$$\|\nabla_{uv}^2\mathbf{h}_i^{(1)}\| \le (c_\sigma\mathsf{w})(\mathsf{w}(\mathbf{S})_{iv}(\mathbf{S})_{iu}) + \mathsf{w}c^{(2)}c_\sigma(\delta_{iv}\mathsf{A}_{iu} + \delta_{iu}\mathsf{A}_{iv} + \sum_j \delta_{jv}(\mathrm{diag}(\mathbf{A}\mathbf{1}) + \mathbf{A})_{ij}\delta_{ju})$$
$$= (c_\sigma\mathsf{w})(\mathsf{w}(\mathbf{S})_{iv}(\mathbf{S})_{iu}) + c^{(2)}c_\sigma\mathsf{w}\mathsf{P}_{i(vu)}^{(0)},$$

which proves the bound for the second-order derivatives in the case $m = 1$.

We now assume that the claim holds for all layers $t \leq m - 1$, and compute the second order derivative after $m$ layers:

$$\left(\nabla_{uv}^2 \mathbf{h}_i^{(m)}\right)_{\alpha, d(\beta-1)+\gamma} = \sigma''(\tilde{h}_i^{(m-1),\alpha})$$

$$\times \left( \sum_r \Omega_{\alpha r}^{(m)}(\nabla_u \mathbf{h}_i^{(m-1)})_{r\beta} \right.$$

$$+ W_{\alpha r}^{(m)} \sum_j \mathsf{A}_{ij} \left( \sum_p \partial_{1,p}\psi^{(m),r}(\nabla_u \mathbf{h}_i^{(m-1)})_{p\beta} + \partial_{2,p}\psi^{(m),r}(\nabla_u \mathbf{h}_j^{(m-1)})_{p\beta}\right) \right)$$

$$\times \left( \sum_r \Omega_{\alpha r}^{(m)}(\nabla_v \mathbf{h}_i^{(m-1)})_{r\gamma} \right.$$

$$+ W_{\alpha r}^{(m)} \sum_j \mathsf{A}_{ij} \left( \sum_p \partial_{1,p}\psi^{(m),r}(\nabla_v \mathbf{h}_i^{(m-1)})_{p\gamma} + \partial_{2,p}\psi^{(m),r}(\nabla_v \mathbf{h}_j^{(m-1)})_{p\gamma}\right) \right)$$

$$+ \sigma'(\tilde{h}_i^{(m-1),\alpha}) \sum_r W_{\alpha r}^{(m)} \sum_j \mathsf{A}_{ij} \sum_{p,q} \partial_{1,p}\partial_{1,q}\psi^{(m),r}(\nabla_u \mathbf{h}_i^{(m-1)})_{p\beta}(\nabla_v \mathbf{h}_i^{(m-1)})_{q\gamma}$$

$$+ \sigma'(\tilde{h}_i^{(m-1),\alpha}) \sum_r W_{\alpha r}^{(m)} \sum_j \mathsf{A}_{ij} \sum_{p,q} \partial_{1,p}\partial_{2,q}\psi^{(m),r}(\nabla_u \mathbf{h}_i^{(m-1)})_{p\beta}(\nabla_v \mathbf{h}_j^{(m-1)})_{q\gamma}$$

$$+ \sigma'(\tilde{h}_i^{(m-1),\alpha}) \sum_r W_{\alpha r}^{(m)} \sum_j \mathsf{A}_{ij} \sum_{p,q} \partial_{1,p}\partial_{2,q}\psi^{(m),r}(\nabla_u \mathbf{h}_j^{(m-1)})_{q\beta}(\nabla_v \mathbf{h}_i^{(m-1)})_{p\gamma}$$

$$+ \sigma'(\tilde{h}_i^{(m-1),\alpha}) \sum_r W_{\alpha r}^{(m)} \sum_j \mathsf{A}_{ij} \sum_{p,q} \partial_{2,p}\partial_{2,q}\psi^{(m),r}(\nabla_u \mathbf{h}_j^{(m-1)})_{p\beta}(\nabla_v \mathbf{h}_j^{(m-1)})_{q\gamma}$$

$$+ \sigma'(\tilde{h}_i^{(m-1),\alpha}) \sum_r \Omega_{\alpha r}^{(m)}(\nabla_{uv}^2 \mathbf{h}_i^{(m-1)})_{r, d(\beta-1)+\gamma}$$

$$+ \sigma'(\tilde{h}_i^{(m-1),\alpha}) \sum_r W_{\alpha r}^{(m)} \sum_j \mathsf{A}_{ij} \left( \sum_p \partial_{1,p}\psi^{(m),r}(\nabla_{uv}^2 \mathbf{h}_i^{(m-1)})_{p, d(\beta-1)+\gamma} \right)$$

$$+ \sigma'(\tilde{h}_i^{(m-1),\alpha}) \sum_r W_{\alpha r}^{(m)} \sum_j \mathsf{A}_{ij} \left( \sum_p \partial_{2,p}\psi^{(m),r}(\nabla_{uv}^2 \mathbf{h}_j^{(m-1)})_{p, d(\beta-1)+\gamma} \right)$$

$$:= R_{\alpha,\beta,\gamma} + \sum_{a,b \in \{1,2\}} (Q_{ab})_{\alpha,\beta,\gamma} + Z_{\alpha,\beta,\gamma},$$

where $R$ is the term containing second derivatives of the non-linear map $\sigma$, $Q_{ab}$ is indexed according to the second derivatives of the message-functions $\psi$, and finally $Z$ is the term containing second-order derivatives of the features. For the term $R_{\alpha,\beta,\gamma}$ we can argue as in the $m = 1$ case and use the sub-matrix notation in (18) to rewrite it as the entry $(\alpha, d(\beta - 1) + \gamma)$ of the $d \times (d \times d)$ sub-matrix

$$\mathbf{R}_{\alpha, d(\beta-1)+\gamma} = (\mathbf{B} \otimes \mathbf{C})'_{\alpha, d(\beta-1)+\gamma}, \tag{22}$$

where

$$\mathbf{B} := \operatorname{diag}(\sigma''(\tilde{\mathbf{h}}_i^{(m-1)}))(\boldsymbol{\Omega}^{(m)}\nabla_u \mathbf{h}_i^{(m-1)} + \mathbf{W}^{(m)}(\sum_j \mathsf{A}_{ij}\nabla_1\psi^{(m)}\nabla_u \mathbf{h}_i^{(m-1)} + \nabla_2\psi^{(m)}\nabla_u \mathbf{h}_j^{(m-1)})),$$

$$\mathbf{C} := \boldsymbol{\Omega}^{(m)}\nabla_v \mathbf{h}_i^{(m-1)} + \mathbf{W}^{(m)}(\sum_j \mathsf{A}_{ij}\nabla_1\psi^{(m)}\nabla_v \mathbf{h}_i^{(m-1)} + \nabla_2\psi^{(m)}\nabla_v \mathbf{h}_j^{(m-1)})$$

Next we consider the terms $(Q_{ab})_{\alpha,\beta,\gamma}$. Without loss of generality, we focus on $(Q_{11})_{\alpha,\beta,\gamma}$ and use again the same argument in the $m = 1$ case, to rewrite it as $(Q_{11})_{\alpha,\beta,\gamma} = (\mathbf{Q}_{11})_{\alpha, d(\beta-1)+\gamma}$ where

$$\mathbf{Q}_{11} = \operatorname{diag}(\sigma'(\tilde{\mathbf{h}}_i^{(m-1)})) \sum_j \mathsf{A}_{ij}(\mathbf{W}^{(m)}\nabla_{11}^2\psi^{(m)}\nabla_u \mathbf{h}_i^{(m-1)} \otimes \nabla_v \mathbf{h}_i^{(m-1)}), \tag{23}$$

where again we are indexing the matrix $\nabla_{11}^2\psi^{(m)}$ by

$$(\nabla_{11}^2\psi^{(m)})_{r,p(d-1)+q} = \partial_{1,p}\partial_{1,q}\psi^{(m),r}.$$

The other $Q$-terms can be estimated similarly. Finally, we rewrite $Z_{\alpha,\beta,\gamma} = (\mathbf{Z})_{\alpha,d(\beta-1)+\gamma}$, where

$$\mathbf{Z} = \text{diag}(\sigma'(\tilde{\mathbf{h}}_i^{(m-1)}))\Big(\mathbf{\Omega}^{(m)}\nabla_{uv}^2\mathbf{h}_i^{(m-1)} + \mathbf{W}^{(m)}\sum_j \mathsf{A}_{ij}(\nabla_1\psi^{(m)}\nabla_{uv}^2\mathbf{h}_i^{(m-1)} + \nabla_2\psi^{(m)}\nabla_{uv}^2\mathbf{h}_j^{(m-1)})\Big) \quad (24)$$

*Therefore, we have rewritten the second-derivatives of the features in matricial form as*

$$\nabla_{uv}^2\mathbf{h}_i^{(m)} = \mathbf{R} + \sum_{a,b\in\{1,2\}}\mathbf{Q}_{ab} + \mathbf{Z}.$$

To complete the proof, we now simply need to estimate the three terms and show they fit the recursion claimed for $m$. For the case of $\mathbf{R}$ in (22), we find

$$\|\mathbf{R}\| \le c_\sigma(\omega\|\nabla_u\mathbf{h}_i^{(m-1)}\| + \mathsf{w}(c_1\text{diag}(\mathbf{A}\mathbf{1})_i\|\nabla_u\mathbf{h}_i^{(m-1)}\| + c_2\sum_j\mathsf{A}_{ij}\|\nabla_u\mathbf{h}_j^{(m-1)}\|))$$

$$\times(\omega\|\nabla_v\mathbf{h}_i^{(m-1)}\| + \mathsf{w}(c_1\text{diag}(\mathbf{A}\mathbf{1})_i\|\nabla_v\mathbf{h}_i^{(m-1)}\| + c_2\sum_j\mathsf{A}_{ij}\|\nabla_v\mathbf{h}_j^{(m-1)}\|)).$$

If we write $D\mathbf{h}^{(m-1)} \in \mathbb{R}^{n\times n}$ as the matrix with entries $(D\mathbf{h}^{(m-1)})_{ij} = \|\nabla_j\mathbf{h}_i^{(m-1)}\|$, then we obtain

$$\|\mathbf{R}\| \le c_\sigma\mathsf{w}(\mathsf{w}\mathbf{S}D\mathbf{h}^{(m-1)})_{iv}(\mathbf{S}D\mathbf{h}^{(m-1)})_{iu}.$$

We can then plug the first-order estimates derived in Theorem C.1 and obtain

$$\|\mathbf{R}\| \le c_\sigma\mathsf{w}(\mathsf{w}\mathbf{S}(c_\sigma\mathsf{w})^{m-1}\mathbf{S}^{m-1})_{iv}(\mathbf{S}(c_\sigma\mathsf{w})^{m-1}\mathbf{S}^{m-1})_{iu} = (c_\sigma\mathsf{w})^{2m-1}(\mathsf{w}(\mathbf{S}^m)_{iv}(\mathbf{S}^m)_{iu}). \quad (25)$$

Next, we move onto the $Q$-terms, and use again the first-order estimates in Theorem C.1 – and the fact that we can bound the norm of $\nabla_{ab}^2\psi^{(m)}$ by $c^{(2)}$ – to derive

$$\|\sum_{a,b\in\{1,2\}}\mathbf{Q}_{ab}\| \le c^{(2)}(c_\sigma\mathsf{w})^{2m-1}(\text{diag}(\mathbf{A}\mathbf{1})_i(\mathbf{S}^{m-1})_{iv}(\mathbf{S}^{m-1})_{iu} + \sum_j\mathsf{A}_{ij}(\mathbf{S}^{m-1})_{ju}(\mathbf{S}^{m-1})_{jv})$$

$$+ c^{(2)}(c_\sigma\mathsf{w})^{2m-1}((\mathbf{S}^{m-1})_{iv}(\mathbf{A}\mathbf{S}^{m-1})_{iu} + (\mathbf{S}^{m-1})_{iu}(\mathbf{A}\mathbf{S}^{m-1})_{iv})$$

$$= c^{(2)}(c_\sigma\mathsf{w})^{2m-1}\mathsf{P}_{i(vu)}^{(m-1)}. \quad (26)$$

Finally, if we let $D^2\mathbf{h}_{vu} \in \mathbb{R}^n$ be the vector with entries $(D^2\mathbf{h}_{vu})_i = \|\nabla_{uv}^2\mathbf{h}_i^{(m-1)}\|$, then

$$\|\mathbf{Z}\| \le c_\sigma\Big(\omega\|\nabla_{uv}^2\mathbf{h}_i^{(m-1)}\| + \mathsf{w}\Big(c_1\text{diag}(\mathbf{A}\mathbf{1})_i\|\nabla_{uv}^2\mathbf{h}_i^{(m-1)}\| + c_2\sum_j\mathsf{A}_{ij}\|\nabla_{uv}^2\mathbf{h}_j^{(m-1)}\|\Big)\Big) \quad (27)$$

$$= c_\sigma\mathsf{w}(\mathbf{S}D^2\mathbf{h}_{vu})_i.$$

Therefore, we can use the induction to derive

$$\|\mathbf{Z}\| \le c_\sigma\mathsf{w}\sum_s\mathbf{S}_{is}\sum_{k=0}^{m-2}\sum_{j\in\mathsf{V}}(c_\sigma\mathsf{w})^{2m-2-k-1}\mathsf{w}(\mathbf{S}^{m-1-k})_{jv}(\mathbf{S}^k)_{sj}(\mathbf{S}^{m-1-k})_{ju}$$

$$+ c_\sigma\mathsf{w}\sum_s\mathbf{S}_{is}(c^{(2)}\sum_{\ell=0}^{m-2}(c_\sigma\mathsf{w})^{m-1+\ell}(\mathbf{S}^{m-2-\ell}\mathsf{P}_{(vu)}^{(\ell)})_s)$$

$$= \sum_{k=0}^{m-2}\sum_{j\in\mathsf{V}}(c_\sigma\mathsf{w})^{2m-k-2}\mathsf{w}(\mathbf{S}^{m-1-k})_{jv}(\mathbf{S}^{k+1})_{ij}(\mathbf{S}^{m-1-k})_{ju}$$

$$+ c^{(2)}(\sum_{\ell=0}^{m-2}(c_\sigma\mathsf{w})^{m+\ell}(\mathbf{S}^{m-1-\ell}\mathsf{P}_{(vu)}^{(\ell)})_i)$$

$$= \sum_{k=1}^{m-1}\sum_{j\in\mathsf{V}}(c_\sigma\mathsf{w})^{2m-k-1}\mathsf{w}(\mathbf{S}^{m-k})_{jv}(\mathbf{S}^k)_{ij}(\mathbf{S}^{m-k})_{ju} + c^{(2)}(\sum_{\ell=0}^{m-2}(c_\sigma\mathsf{w})^{m+\ell}(\mathbf{S}^{m-1-\ell}\mathsf{P}_{(vu)}^{(\ell)})_i)$$

By (25), we derive that the $\mathbf{R}$-term corresponds to the $k = 0$ entry of the first sum, while (26) corresponds to the case $\ell = m - 1$ of the second sum, which completes the induction and hence our proof. $\qquad\square$

### C.1 Proof of Theorem 3.2

We can now use the previous characterization to derive estimates on the Hessian of the graph-level function computed by MPNNs. We restate Theorem 3.2 here for convenience.

**Theorem 3.2.** *Consider an* MPNN *of depth $m$ as in* (2)*, where $\sigma$ and $\psi^{(t)}$ are $\mathcal{C}^2$ functions and we denote the bounds on their derivatives and on the norm of the weights as above. Let $\mathbf{S}$ and $\mathbf{Q}_k$ be defined as in* (5) *and* (6)*, respectively. If the readout is* MAX, MEAN *or* SUM *and $\boldsymbol{\theta}$ in* (3) *has unit norm, then the mixing* $\mathsf{mix}_{y_{\mathsf{G}}^{(m)}}(v, u)$ *induced by the* MPNN *over the features of nodes $v, u$ satisfies*

$$\mathsf{mix}_{y_{\mathsf{G}}^{(m)}}(v, u) \le \sum_{k=0}^{m-1} (c_\sigma \mathsf{w})^{2m-k-1} \left( \mathsf{w}(\mathbf{S}^{m-k})^\top \mathrm{diag}(\mathbf{1}^\top \mathbf{S}^k) \mathbf{S}^{m-k} + c^{(2)} \mathbf{Q}_k \right)_{vu}. \tag{7}$$

*Proof.* First, we recall that according to Definition 3.1, we are interested in bounding the quantity

$$\mathsf{mix}_{y_{\mathsf{G}}^{(m)}}(v, u) = \max_{\mathbf{X}} \max_{1 \le \beta, \gamma \le d} \left| \frac{\partial^2 y_{\mathsf{G}}^{(m)}(\mathbf{X})}{\partial x_u^\beta \partial x_v^\gamma} \right|.$$

Let us first consider the choice READ = SUM, so that by (3) we get

$$\mathsf{mix}_{y_{\mathsf{G}}^{(m)}}(v, u) \le \left| \sum_{\alpha=1}^{d} \theta_\alpha \sum_{i \in \mathsf{V}} \frac{\partial^2 h_i^{(m),\alpha}}{\partial x_u^\beta \partial x_v^\gamma} \right|$$

As before, we index the columns of the Hessian of $\mathbf{h}_i$ as $\frac{\partial^2 h_i^{(m),\alpha}}{\partial x_u^\beta \partial x_v^\gamma} = (\nabla_{uv}^2 \mathbf{h}_i^{(m)})_{\alpha, d(\beta-1)+\gamma}$ and hence obtain

$$\mathsf{mix}_{y_{\mathsf{G}}^{(m)}}(v, u) \le \sum_{i \in \mathsf{V}} \|(\nabla_{uv}^2 \mathbf{h}_i^{(m)})^\top \boldsymbol{\theta}\| \le \sum_{i \in \mathsf{V}} \|\nabla_{uv}^2 \mathbf{h}_i^{(m)}\|, \tag{28}$$

since $\boldsymbol{\theta}$ has unit norm. *Note that the very same bound in* (28) *also holds if we replaced the* SUM *readout with either the* MAX *or the* MEAN *readout.* We can then rely on Theorem C.2 and find

$$\begin{aligned}
\mathsf{mix}_{y_{\mathsf{G}}^{(m)}}(v, u) &\le \sum_{i \in \mathsf{V}} \sum_{k=0}^{m-1} \sum_{j \in \mathsf{V}} (c_\sigma \mathsf{w})^{2m-k-1} \mathsf{w}(\mathbf{S}^{m-k})_{jv} (\mathbf{S}^k)_{ij} (\mathbf{S}^{m-k})_{ju} \\
&\quad + c^{(2)} \sum_{i \in \mathsf{V}} \sum_{\ell=0}^{m-1} (c_\sigma \mathsf{w})^{m+\ell} (\mathbf{S}^{m-1-\ell} \mathsf{P}_{(vu)}^{(\ell)})_i \\
&= \sum_{k=0}^{m-1} (c_\sigma \mathsf{w})^{2m-k-1} \left( \mathsf{w}(\mathbf{S}^{m-k})^\top \mathrm{diag}(\mathbf{1}^\top \mathbf{S}^k) \mathbf{S}^{m-k} \right)_{vu} \\
&\quad + c^{(2)} \sum_{\ell=0}^{m-1} (c_\sigma \mathsf{w})^{m+\ell} (\mathbf{1}^\top \mathbf{S}^{m-1-\ell}) \mathsf{P}_{(vu)}^{(\ell)} \\
&= \sum_{k=0}^{m-1} (c_\sigma \mathsf{w})^{2m-k-1} \left( \mathsf{w}(\mathbf{S}^{m-k})^\top \mathrm{diag}(\mathbf{1}^\top \mathbf{S}^k) \mathbf{S}^{m-k} \right)_{vu} \\
&\quad + c^{(2)} \sum_{k=0}^{m-1} (c_\sigma \mathsf{w})^{2m-k-1} (\mathbf{1}^\top \mathbf{S}^k) \mathsf{P}_{(vu)}^{(m-k-1)}.
\end{aligned}$$

For the second term we can the simply use the formula in (15) to rewrite it in matricial form as claimed – recall that $\mathbf{Q}_k$ is defined in (6). $\qquad\square$

# D   Proofs and additional details of Section 4

Throughout this Section, for simplicity, we assume $c_\sigma = \max\{|c'_\sigma|, |c''_\sigma|\}$ to be smaller or equal than one – this is satisfied by the vast majority of commonly used non-linear activations, and extending the results below to arbitrary $c_\sigma$ is straightforward.

## D.1   Proof of Theorem 4.3

We begin by proving lower bounds on the operator norm of the weights, when the depth is the minimal one required to induce any non-zero mixing among nodes $v, u$. For convenience, we restate Theorem 4.3 here as well.

**Theorem 4.3.** *Let* $\mathbf{A} = \mathbf{A}_{\text{sym}}$, $r := d_{\mathsf{G}}(v, u)$, $m = \lceil r/2 \rceil$, *and* $q$ *be the number of paths of length* $r$ *between* $v$ *and* $u$. *For an* MPNN *satisfying Theorem 3.2 with capacity* $(m = \lceil r/2 \rceil, \mathsf{w})$, *we find* $\widetilde{\mathsf{OSQ}}_{v,u}(m, \mathsf{w}) \cdot (c_2 \mathsf{w})^r (\mathbf{A}^r)_{vu} \geq 1$. *In particular, if the* MPNN *generates mixing* $\mathsf{mix}_{y_{\mathsf{G}}}(v, u)$, *then*

$$\mathsf{w} \geq \frac{d_{\min}}{c_2} \left( \frac{\mathsf{mix}_{y_{\mathsf{G}}}(v, u)}{q} \right)^{\frac{1}{r}}.$$

*Proof.* Without loss of generality, we assume that $r$ is even, so that by our assumptions, we can simply take $m = r/2$. According to Theorem 3.2, we know that the maximal mixing induced by an MPNN of depth $m$ over the features associated with nodes $v, u$ is bounded from above as

$$\mathsf{mix}_{y_{\mathsf{G}}^{(m)}}(v, u) \leq \sum_{k=0}^{m-1} \mathsf{w}^{2m-k-1} \left( \mathsf{w} \mathbf{S}^{m-k} \mathrm{diag}(\mathbf{S}^k \mathbf{1}) \mathbf{S}^{m-k} + c^{(2)} \mathbf{Q}_k \right)_{vu}.$$

where we have replaced $c_\sigma$ with one, as per our assumption. Since $m = r/2$, where $r$ is the distance among nodes $v, u$, then the only non-zero contribution for the first term is obtained for $k = 0$ – otherwise we would find a path of length $2(m - k)$ connecting $v$ and $u$ hence violating the assumptions – and is equal to $\mathsf{w}^{2m}(\mathbf{S}^{2m})_{vu}$, and note that $2m = r$. Concerning the terms $\mathbf{Q}_k$ instead, the longest-walk contribution for nodes $v, u$ is $2m - 1$ (when $k = 0$), meaning that $\mathbf{Q}_k = 0$ for all $0 \leq k \leq m - 1$ if $m = 2r$. Accordingly, we can reduce the bound above to:

$$\mathsf{mix}_{y_{\mathsf{G}}^{(m)}}(v, u) \leq \mathsf{w}^{2m} \left( \mathbf{S}^{2m} \right)_{vu} = \mathsf{w}^{2m} \left( (c_2 \mathbf{A})^{2m} \right)_{vu},$$

where in the last equality we have again used that $2m = r$, so when expanding the power of $\mathbf{S}$ only the highest-order term in the $\mathbf{A}$-variable gives non-zero contributions. If we replace now $2m = r$ and use the characterization of over-squashing in Definition 4.2, then

$$\widetilde{\mathsf{OSQ}}_{v,u}(m = \frac{r}{2}, \mathsf{w}) \geq (c_2 \mathsf{w})^{-r} \frac{1}{(\mathbf{A}^r)_{vu}}.$$

Therefore, if the MPNN generates mixing $\mathsf{mix}_{y_{\mathsf{G}}^{(m)}}(v, u)$ among the features of $v$ and $u$, then (8) is satisfied, meaning that the operator norm of the weights must be larger than

$$\mathsf{w} \geq \frac{1}{c_2} \left( \frac{\mathsf{mix}_{y_{\mathsf{G}}^{(m)}}(v, u)}{(\mathbf{A}^r)_{vu}} \right)^{\frac{1}{r}}.$$

The term $\mathbf{A}^r$ in general can be estimated sharply depending on the knowledge we have of the underlying graph. To get a universal – albeit potentially looser bound – it suffices to note that along each path connecting $v$ and $u$, the product of the entries of $\mathbf{A}$ can be bounded from above by $(d_{\min})^r$, which completes the proof.   $\square$

We highlight that if $\mathbf{A} = \mathbf{A}_{\text{rw}} = \mathbf{D}^{-1}\mathbf{A}$ – i.e. the aggregation over the neighbours consists of a mean-operation as for the GraphSAGE architecture – then one can apply the very same proof above and derive

**Corollary D.1.** *The same lower bound for* $\mathsf{w}$ *in Theorem 4.3, holds when* $\mathbf{A} = \mathbf{D}^{-1}\mathbf{A}$.

### D.2 Spectral bounds

Next, we study the case of fixed, bounded operator norm of the weights, but variable depth, since we are interested in showing that *over-squashing hinders the expressive power of MPNNs for tasks requiring high-mixing of features associated with nodes at high commute time.* We first provide a characterization of the maximal mixing (and hence of the over-squashing measure) in terms of the graph-Laplacian and its pseudo-inverse.

**Convention.** In the proofs below we usually deal with matrices with nonnegative entries. Accordingly, we introduce the following convention: we write that $\mathbf{A} \leq \mathbf{B}$ if $A_{ij} \leq B_{ij}$ for all entries $1 \leq i, j \leq n$.

**Theorem D.2.** *Let $\gamma := \sqrt{\frac{d_{\max}}{d_{\min}}}$ and set $\mathbf{A} = \mathbf{A}_{\mathrm{sym}}$ or $\mathbf{A} = \mathbf{A}_{\mathrm{rw}}$. Consider an MPNN as in Thm. 3.2 with depth $m$, $\max\{\mathsf{w}, \omega/\mathsf{w} + c_1\gamma + c_2\} \leq 1$. Define $\mathbf{Z} := \mathbf{I} - c_2\mathbf{\Delta}$. Then the maximal mixing of nodes $v, u$ generated by such MPNN after $m$ layers is*

$$\mathrm{mix}_{y_{\mathsf{G}}^{(m)}}(v, u) \leq \gamma^k \Big( m \frac{\sqrt{d_v d_u}}{2|\mathsf{E}|} \big(1 + 2c^{(2)}(1 + \gamma^s)\big) + \frac{1}{c_2} \Big(\mathbf{Z}^2(\mathbf{I} - \mathbf{Z}^{2m})(\mathbf{I} + \mathbf{Z})^{-1}\mathbf{\Delta}^\dagger\Big)_{vu}\Big)$$

$$+ 2\frac{c^{(2)}}{c_2}\gamma^k \Big(\big((1 + \gamma^s)\mathbf{I} - \mathbf{\Delta}\big)(\mathbf{I} - \mathbf{Z}^{2m})(\mathbf{I} + \mathbf{Z})^{-1}\mathbf{\Delta}^\dagger\Big)_{vu},$$

*where $k = s = 1$ if $\mathbf{A} = \mathbf{A}_{\mathrm{sym}}$ or $k = 4, s = 2$ if $\mathbf{A} = \mathbf{A}_{\mathrm{rw}}$.*

*Proof.* We first focus on the symmetrically normalized case $\mathbf{A} = \mathbf{A}_{\mathrm{sym}} = \mathbf{D}^{-1/2}\mathbf{A}\mathbf{D}^{-1/2}$, which we recall that we can rewrite as $\mathbf{A}_{\mathrm{sym}} = \mathbf{I} - \mathbf{\Delta}$, where $\mathbf{\Delta}$ is the (normalized) graph Laplacian (10). Since $\mathsf{w} \leq 1$ and the message-passing matrix is symmetric, by Theorem 3.2, we can bound the maximal mixing of an MPNN as in the statement by

$$\mathrm{mix}_{y_{\mathsf{G}}^{(m)}}(v, u) \leq \sum_{k=0}^{m-1} \Big(\mathbf{S}^{m-k}\mathrm{diag}(\mathbf{S}^k \mathbf{1})\mathbf{S}^{m-k} + c^{(2)}\mathbf{Q}_k\Big)_{vu}.$$

We focus on the first sum. Since $\mathbf{A} = \mathbf{D}^{1/2}(\mathbf{D}^{-1}\mathbf{A})\mathbf{D}^{-1/2}$, where $\mathbf{D}^{-1}\mathbf{A}$ is a row-stochastic matrix, we see that

$$\mathsf{S}_{ij} \leq \frac{\omega}{\mathsf{w}}\delta_{ij} + c_1\gamma\delta_{ij} + c_2\mathsf{A}_{ij},$$

meaning that we can write $\mathbf{S} \leq \alpha\mathbf{I} + c_2\mathbf{A}$, where $\alpha = \omega/\mathsf{w} + c_1\gamma$, using the convention introduced above. Accordingly, we can estimate the row-sum of the powers of $\mathbf{S}$ by using $(\mathbf{A}^p \mathbf{1})_i \leq \gamma$ as

$$(\mathbf{S}^k \mathbf{1})_i \leq \sum_{p=0}^{k} \binom{k}{p}\alpha^{k-p}c_2^p(\mathbf{A}^p \mathbf{1})_i \leq \gamma\sum_{p=0}^{k} \binom{k}{p}\alpha^{k-p}c_2^p = \gamma(\alpha + c_2)^k \leq \gamma,$$

where the last inequality simply follows from the assumptions. Therefore, we find

$$\Big(\sum_{k=0}^{m-1} \mathbf{S}^{m-k}\mathrm{diag}(\mathbf{S}^k \mathbf{1})\mathbf{S}^{m-k}\Big)_{vu} \leq \gamma\sum_{k=0}^{m-1}(\mathbf{S}^{2(m-k)})_{vu} = \gamma\sum_{k=1}^{m}(\mathbf{S}^{2k})_{vu}. \tag{29}$$

By the assumptions on the regularity of the message-functions, we can estimate $\mathbf{S}$ from above by $\mathbf{S} \leq \alpha\mathbf{I} + c_2\mathbf{A} = (\alpha + c_2)\mathbf{I} - c_2\mathbf{\Delta} \leq \mathbf{Z}$, and derive

$$\Big(\sum_{k=0}^{m-1} \mathbf{S}^{m-k}\mathrm{diag}(\mathbf{S}^k \mathbf{1})\mathbf{S}^{m-k}\Big)_{vu} \leq \gamma\sum_{k=1}^{m}(\mathbf{Z}^{2k})_{vu}.$$

From the spectral decomposition of the graph-Laplacian in (10) and the properties that $\lambda_0 = 0$ and $\phi_0(v) = \sqrt{d_v/2|\mathsf{E}|}$, we find

$$
\begin{aligned}
\sum_{k=1}^{m}(\mathbf{Z}^{2k})_{vu} &= \sum_{k=1}^{m}\sum_{\ell=0}^{n-1}(1-c_2\lambda_\ell)^{2k}\phi_\ell(v)\phi_\ell(u) \\
&= m\frac{\sqrt{d_v d_u}}{2|\mathsf{E}|} + \sum_{\ell=1}^{n-1}\Big(\frac{1-(1-c_2\lambda_\ell)^{2(m+1)}}{1-(1-c_2\lambda_\ell)^2}-1\Big)\phi_\ell(v)\phi_\ell(u) \\
&= m\frac{\sqrt{d_v d_u}}{2|\mathsf{E}|} + \sum_{\ell=1}^{n-1}\Big(\frac{(1-c_2\lambda_\ell)^2(1-(1-c_2\lambda_\ell)^{2m})}{(2-c_2\lambda_\ell)c_2\lambda_\ell}\Big)\phi_\ell(v)\phi_\ell(u).
\end{aligned}
$$

Since $c_2 \leq 1$ and $\mathsf{G}$ is not bipartite, we derive that $(\mathbf{I} + \mathbf{Z}) = 2\mathbf{I} - c_2\boldsymbol{\Delta}$ is invertible and hence that the following decomposition holds:

$$
(\mathbf{I} + \mathbf{Z})^{-1} = \sum_{\ell \geq 0}\frac{1}{2-c_2\lambda_\ell}\phi_\ell\phi_\ell^\top.
$$

Therefore, we can rely on the spectral-decomposition of the pseudo-inverse of the graph-Laplacian in (11) to get

$$
\Big(\sum_{k=0}^{m-1}\mathbf{S}^{m-k}\mathrm{diag}(\mathbf{S}^k\mathbf{1})\mathbf{S}^{m-k}\Big)_{vu} \leq \gamma\Big(m\frac{\sqrt{d_v d_u}}{2|\mathsf{E}|}+\frac{1}{c_2}\Big(\mathbf{Z}^2(\mathbf{I}-\mathbf{Z}^{2m})(\mathbf{I}+\mathbf{Z})^{-1}\boldsymbol{\Delta}^\dagger\Big)_{vu}\Big). \tag{30}
$$

It now remains to bound the term $c^{(2)}\sum_{k=0}^{m-1}(\mathbf{Q}_k)_{vu}$. First, we note that by the symmetry of $\mathbf{A}$ and the estimate $(\mathbf{S}^k\mathbf{1})_i \leq \gamma$, that we derived above, we obtain

$$
c^{(2)}\sum_{k=0}^{m-1}(\mathbf{Q}_k)_{vu} \leq 2c^{(2)}\gamma\Big(\sum_{k=0}^{m-1}(\mathbf{A}\mathbf{Z}^{2(m-k-1)})_{vu}+\gamma(\mathbf{Z}^{2(m-k-1)})_{vu}\Big).
$$

Then we can use the identity $\mathbf{A} = \mathbf{I} - \boldsymbol{\Delta}$, to find

$$
c^{(2)}\sum_{k=0}^{m-1}(\mathbf{Q}_k)_{vu} \leq 2c^{(2)}\gamma\sum_{k=0}^{m-1}\Big(((1+\gamma)\mathbf{I}-\boldsymbol{\Delta})\mathbf{Z}^{2(m-k-1)}\Big)_{vu}. \tag{31}
$$

By relying on the spectral decomposition as above, we finally get

$$
\begin{aligned}
c^{(2)}\sum_{k=0}^{m-1}(\mathbf{Q}_k)_{vu} \leq{}& 2c^{(2)}\gamma\Big(m\frac{\sqrt{d_v d_u}}{2|\mathsf{E}|}(1+\gamma)\Big) \\
&+ 2c^{(2)}\gamma\Big(\sum_{\ell>0}(1+\gamma-\lambda_\ell)\frac{1-(1-c_2\lambda_\ell)^{2m}}{(2-c_2\lambda_\ell)c_2\lambda_\ell}\phi_\ell(v)\phi_\ell(u)\Big).
\end{aligned}
$$

As done previously, we can rewrite the last terms via $(\mathbf{I} + \mathbf{Z})^{-1}$ as

$$
\begin{aligned}
c^{(2)}\sum_{k=0}^{m-1}(\mathbf{Q}_k)_{vu} \leq{}& 2c^{(2)}\gamma\Big(m\frac{\sqrt{d_v d_u}}{2|\mathsf{E}|}(1+\gamma)\Big) \\
&+ 2\frac{c^{(2)}}{c_2}\gamma\Big((1+\gamma)\mathbf{I}-\boldsymbol{\Delta})(\mathbf{I}-\mathbf{Z}^{2m})(\mathbf{I}+\mathbf{Z})^{-1}\boldsymbol{\Delta}^\dagger\Big)_{vu}. 
\end{aligned} \tag{32}
$$

We can then combine (30) and (32) and derive the bound we claimed, when $\mathbf{A} = \mathbf{A}_{\mathrm{sym}}$. For the case $\mathbf{A} = \mathbf{A}_{\mathrm{rw}} = \mathbf{D}^{-1}\mathbf{A}$, it suffices to notice that $\mathbf{S} \leq \alpha'\mathbf{I} + c_2\mathbf{A}$, where $\alpha' = \omega/\mathsf{w} + c_1$ and that

$$
(\mathbf{1}^\top\mathbf{S}^k)_i \leq \sum_{j}\sum_{p=0}^{k}\binom{k}{p}(\alpha')^{k-p}c_2^p((\mathbf{D}^{-1}\mathbf{A})^p)_{ji} \leq \frac{d_{\max}}{d_{\min}}(\alpha'+c_2)^k \leq \gamma^2,
$$

where we have used that by assumption $\alpha' + c_2 \leq 1$. Similarly, we get $(\mathbf{S}^{m-k})^\top \mathbf{S}^{(m-k)} \leq \gamma^2 \mathbf{Z}^{2(m-k)}$. Finally, the $\mathbf{Q}_k$-term can be bounded by

$$c^{(2)} \sum_{k=0}^{m-1} (\mathbf{Q}_k)_{vu} \leq 2c^{(2)} \Big( \sum_{k=0}^{m-1} \gamma^4 \mathbf{A}_{\mathrm{sym}} \mathbf{Z}^{2(m-k-1)} + \gamma^6 \mathbf{Z}^{2(m-k-1)} \Big),$$

and we can follow the previous steps in the symmetric case to complete the proof. $\qquad\square$

**Corollary D.3.** *Under the assumptions of Theorem D.2, if the message functions in* (2) *are linear – as for GCN, SAGE, or GIN – then the maximal mixing induced by such an MPNN of $m$ layers is*

$$\mathsf{mix}_{y_{\mathsf{G}}^{(m)}}(v, u) \leq \Big( \frac{d_{\max}}{d_{\min}} \Big)^k \Big( m \frac{\sqrt{d_v d_u}}{2|\mathsf{E}|} + \frac{1}{c_2} \Big( \mathbf{Z}^2 (\mathbf{I} - \mathbf{Z}^{2m})(\mathbf{I} + \mathbf{Z})^{-1} \mathbf{\Delta}^\dagger \Big)_{vu} \Big)$$

*Proof.* This follows from Theorem D.2 simply by noticing that if the message-function $\psi$ in (2) is linear, then the upper bound for the norm of the Hessian can be taken to be zero, i.e. $c^{(2)} = 0$. $\qquad\square$

### D.3 Proof of Theorem 4.4

We now expand the previous results to derive the minimal number of layers required to induce mixing in the case of bounded weights, showing that the depth may need to grow with the commute time of nodes. We recall that $\gamma$ is $\sqrt{d_{\max}/d_{\min}}$ while $0 = \lambda_0 < \lambda_1 \leq \ldots \leq \lambda_{n-1}$ are the eigenvalues of the symmetrically normalized graph Laplacian (10). We restate Theorem 4.4 below.

**Theorem 4.4.** *Consider an* MPNN *satisfying Theorem 3.2, with* $\max\{\mathsf{w}, \omega/\mathsf{w} + c_1\gamma + c_2\} \leq 1$*, and* $\mathbf{A} = \mathbf{A}_{\mathrm{sym}}$*. If* $\widetilde{\mathsf{OSQ}}_{v,u}(m, \mathsf{w}) \cdot (\mathsf{mix}_{y_{\mathsf{G}}}(v, u)) \leq 1$*, i.e. the* MPNN *generates mixing* $\mathsf{mix}_{y_{\mathsf{G}}}(v, u)$ *among the features associated with nodes* $v, u$*, then the number of layers $m$ satisfies*

$$m \geq \frac{\tau(v, u)}{4c_2} + \frac{|\mathsf{E}|}{\sqrt{d_v d_u}} \Big( \frac{\mathsf{mix}_{y_{\mathsf{G}}}(v, u)}{\gamma \mu} - \frac{1}{c_2} \Big( \frac{\gamma + |1 - c_2 \lambda^*|^{r-1}}{\lambda_1} + 2 \frac{c^{(2)}}{\mu} \Big) \Big), \tag{9}$$

*where* $r = d_{\mathsf{G}}(v, u)$*,* $\mu = 1 + 2c^{(2)}(1 + \gamma)$ *and* $|1 - c_2 \lambda^*| = \max_{0 < \ell \leq n-1} |1 - c_2 \lambda_\ell| < 1$*.*

*Proof.* From now on we let $r$ be the shortest-walk distance between $v$ and $u$. If $m < r/2$, then we incur the under-reaching issue and hence we get zero mixing among the features associated with nodes $v, u$. Accordingly, we can choose $m \geq r/2$. We need to provide an estimate on the maximal mixing induced by an MPNN as in the statement. We focus on the bound in Theorem 3.2, and recall that the first sum can be bounded as in (29) by

$$\Big( \sum_{k=0}^{m-1} \mathbf{S}^{m-k} \mathrm{diag}(\mathbf{S}^k \mathbf{1}) \mathbf{S}^{m-k} \Big)_{vu} \leq \gamma \sum_{k=1}^{m} (\mathbf{S}^{2k})_{vu} \leq \gamma \sum_{k=1}^{m} (\mathbf{Z}^{2k})_{vu},$$

where $\mathbf{Z} := \mathbf{I} - c_2 \mathbf{\Delta}$. We can then bound the geometric sum by accounting for the odd powers too. Therefore, we get

$$\gamma \sum_{k=1}^{m} (\mathbf{Z}^{2k})_{vu} \leq \gamma \sum_{k=1}^{2m} (\mathbf{Z}^k)_{vu} = \gamma \sum_{k=0}^{2m} \sum_{\ell \geq 0} (1 - c_2 \lambda_\ell)^k \phi_\ell(v) \phi_\ell(u).$$

As for the proof of Theorem D.2, we separate the contribution of the zero-eigenvalue and that of the positive ones, so we find that

$$\sum_{k=0}^{2m} (\mathbf{Z}^k)_{vu} \leq (2m+1) \frac{\sqrt{d_v d_u}}{2|\mathsf{E}|} + \sum_{\ell > 0} \Big( \frac{1 - (1 - c_2 \lambda_\ell)^{2m+1}}{c_2 \lambda_\ell} \Big) \phi_\ell(v) \phi_\ell(u)$$

$$= (2m+1) \frac{\sqrt{d_v d_u}}{2|\mathsf{E}|} + \sum_{\ell > 0} \frac{1}{c_2 \lambda_\ell} \phi_\ell(v) \phi_\ell(u) - \sum_{\ell > 0} \frac{(1 - c_2 \lambda_\ell)^{2m+1}}{c_2 \lambda_\ell} \phi_\ell(v) \phi_\ell(u). \tag{33}$$

Thanks to the characterization of commute-time provided in (12), we derive

$$\sum_{\ell>0}\frac{1}{c_2\lambda_\ell}\phi_\ell(v)\phi_\ell(u) = -\frac{\tau(v,u)}{4c_2|\mathsf{E}|}\sqrt{d_v d_u} + \frac{1}{2c_2}\sum_{\ell>0}\frac{1}{\lambda_\ell}\left(\phi_\ell^2(v)\sqrt{\frac{d_u}{d_v}} + \phi_\ell^2(u)\sqrt{\frac{d_v}{d_u}}\right)$$

$$\le -\frac{\tau(v,u)}{4c_2|\mathsf{E}|}\sqrt{d_v d_u} + \frac{1}{2c_2\lambda_1}\left(\sqrt{\frac{d_v}{d_u}} + \sqrt{\frac{d_u}{d_v}} - \frac{\sqrt{d_v d_u}}{|\mathsf{E}|}\right) \tag{34}$$

where in the last inequality we have used that $\sum_{\ell>0}\phi_\ell^2(v) = 1 - \phi_0^2(v)$ since $\{\phi_\ell\}$ constitute an orthonormal basis, with $\phi_0(v) = \sqrt{d_v/2|\mathsf{E}|}$, and that $\lambda_\ell \ge \lambda_1$, for all $\ell > 0$. Next, we estimate the second sum in (33), and we note that $\lambda^*$ in the statement is either $\lambda_1$ or $\lambda_{n-1}$:

$$-\sum_{\ell>0}\frac{(1-c_2\lambda_\ell)^{2m+1}}{c_2\lambda_\ell}\phi_\ell(v)\phi_\ell(u) \le \sum_{\ell>0}\frac{|1-c_2\lambda^*|^{2m+1}}{c_2\lambda_\ell}|\phi_\ell(v)\phi_\ell(u)|$$

$$\le \frac{|1-c_2\lambda^*|^{2m+1}}{2c_2\lambda_1}\sum_{\ell>0}(\phi_\ell^2(v) + \phi_\ell^2(u))$$

$$\le \frac{|1-c_2\lambda^*|^r}{2c_2\lambda_1}\left(2 - \frac{d_v}{2|\mathsf{E}|} - \frac{d_u}{2|\mathsf{E}|}\right), \tag{35}$$

where in the last inequality we have used that $|1 - c_2\lambda^*| < 1$ and that $m \ge r/2$ (otherwise we would have zero-mixing due to under-reaching). Therefore, by combining (34) and (35), we derive that the first sum on the right hand side of (7) can be bounded from above by

$$\left(\sum_{k=0}^{m-1}\mathsf{S}^{m-k}\mathrm{diag}(\mathsf{S}^k\mathbf{1})\mathsf{S}^{m-k}\right)_{vu} \le \gamma\left((2m+1)\frac{\sqrt{d_v d_u}}{2|\mathsf{E}|} - \frac{\tau(v,u)}{4c_2|\mathsf{E}|}\sqrt{d_v d_u}\right)$$

$$+ \frac{\gamma}{2c_2\lambda_1}\left(\sqrt{\frac{d_v}{d_u}} + \sqrt{\frac{d_u}{d_v}} - \frac{\sqrt{d_v d_u}}{|\mathsf{E}|}\right)$$

$$+ \gamma\frac{|1-c_2\lambda^*|^r}{2c_2\lambda_1}\left(2 - \frac{d_v}{2|\mathsf{E}|} - \frac{d_u}{2|\mathsf{E}|}\right). \tag{36}$$

Next, we continue by estimating the second sum entering the right hand side of (7). We recall that by (31), we have

$$c^{(2)}\sum_{k=0}^{m-1}(\mathbf{Q}_k)_{vu} \le 2c^{(2)}\gamma\sum_{k=0}^{m-1}\left(((1+\gamma)\mathbf{I} - \boldsymbol{\Delta})\mathbf{Z}^{2(m-k-1)}\right)_{vu} = 2c^{(2)}\gamma\sum_{k=0}^{m-1}\left(((1+\gamma)\mathbf{I} - \boldsymbol{\Delta})\mathbf{Z}^{2k}\right)_{vu}$$

$$\le 2c^{(2)}\gamma\sum_{k=0}^{2(m-1)}\left(((1+\gamma)\mathbf{I} - \boldsymbol{\Delta})\mathbf{Z}^k\right)_{vu}$$

$$= 2c^{(2)}\gamma\sum_{k=0}^{2(m-1)}\sum_{\ell\ge0}((1+\gamma) - \lambda_\ell)(1-c_2\lambda_\ell)^k\phi_\ell(v)\phi_\ell(u).$$

We then proceed as above, and separate the contributions associated with the kernel of the Laplacian, to find

$$\sum_{k=0}^{m-1}(\mathbf{Q}_k)_{vu} \le 2\gamma\left((1+\gamma)(2m-1)\frac{\sqrt{d_v d_u}}{2|\mathsf{E}|}\right)$$

$$+ 2\gamma(1+\gamma)\sum_{\ell>0}\frac{1}{c_2\lambda_\ell}(1 - (1-c_2\lambda_\ell)^{2m-1})\phi_\ell(v)\phi_\ell(u) \tag{37}$$

$$- \frac{2\gamma}{c_2}\sum_{\ell>0}(1 - (1-c_2\lambda_\ell)^{2m-1})\phi_\ell(v)\phi_\ell(u). \tag{38}$$

For the term in (37), we can apply the same estimate as for the case of (33). Similarly, we can bound (38) by

$$-\frac{2\gamma}{c_2}\sum_{\ell>0}(1-(1-c_2\lambda_\ell)^{2m-1})\phi_\ell(v)\phi_\ell(u) \le \frac{2\gamma}{c_2}\sum_{\ell>0}\frac{1}{2}\Big(\phi_\ell^2(v)+\phi_\ell^2(u)\Big) \le \frac{\gamma}{c_2}\Big(2-\frac{d_v}{2|\mathsf{E}|}-\frac{d_u}{2|\mathsf{E}|}\Big).$$

Therefore, we can finally bound the $\mathbf{Q}_k$-terms in (7) by

$$
\begin{aligned}
c^{(2)}\sum_{k=0}^{m-1}(\mathbf{Q}_k)_{vu} \le{}& 2c^{(2)}\gamma\Big((1+\gamma)(2m-1)\frac{\sqrt{d_v d_u}}{2|\mathsf{E}|}-(1+\gamma)\frac{\tau(v,u)}{4c_2|\mathsf{E}|}\sqrt{d_v d_u}\Big)\\
&+c^{(2)}\gamma\Big(\frac{1+\gamma}{c_2\lambda_1}\Big(\sqrt{\frac{d_v}{d_u}}+\sqrt{\frac{d_u}{d_v}}-\frac{\sqrt{d_v d_u}}{|\mathsf{E}|}\Big)\\
&\quad c^{(2)}\gamma\frac{1+\gamma}{c_2\lambda_1}|1-c_2\lambda^*|^{r-1}\Big(2-\frac{d_v}{2|\mathsf{E}|}-\frac{d_u}{2|\mathsf{E}|}\Big)\Big)\\
&+\frac{c^{(2)}\gamma}{c_2}\Big(2-\frac{d_v}{2|\mathsf{E}|}-\frac{d_u}{2|\mathsf{E}|}\Big).
\end{aligned}
\tag{39}
$$

We can the combine (36) and (39), to find that the maximal mixing induced by an MPNN of $m$ layers as in the statement of Theorem 4.4, is

$$
\begin{aligned}
\mathsf{mix}_{y_\mathsf{G}^{(m)}}(v,u) \le{}& \gamma\sqrt{d_v d_u}\Big(\frac{m}{|\mathsf{E}|}\mu+\frac{1}{2|\mathsf{E}|}-\frac{\tau(v,u)}{4c_2|\mathsf{E}|}\mu\Big)\\
&+\gamma\frac{\mu}{2c_2\lambda_1}\Big(\sqrt{\frac{d_v}{d_u}}+\sqrt{\frac{d_u}{d_v}}-\frac{\sqrt{d_v d_u}}{|\mathsf{E}|}\Big)\\
&+\gamma\frac{\mu}{2c_2\lambda_1}|1-c_2\lambda^*|^{r-1}\Big(2-\frac{d_v}{2|\mathsf{E}|}-\frac{d_u}{2|\mathsf{E}|}\Big)\\
&+\frac{\gamma c^{(2)}}{c_2}\Big(2-\frac{d_v}{2|\mathsf{E}|}-\frac{d_u}{2|\mathsf{E}|}\Big),
\end{aligned}
\tag{40}
$$

where $\mu := 1+2c^{(2)}(1+\gamma)$ and we have removed the term $-2c^{(2)}\gamma(1+\gamma)\sqrt{d_v d_u}/2|\mathsf{E}| \le 0$. Moreover, since $\lambda_1 < 1$ unless $\mathsf{G}$ is the complete graph (and if that was the case, then we could take the distance $r$ below to simply be equal to 1) and $c_2 \le 1$, we find

$$\gamma\sqrt{d_v d_u}\frac{1}{2|\mathsf{E}|}\Big(1-\frac{\mu}{c_2\lambda_1}\Big(1+\frac{|1-c_2\lambda^*|^{r-1}}{2}\Big(\sqrt{\frac{d_v}{d_u}}+\sqrt{\frac{d_u}{d_v}}\Big)\Big)\Big) \le 0.$$

Accordingly, we can simplify (40) as

$$
\begin{aligned}
\mathsf{mix}_{y_\mathsf{G}^{(m)}}(v,u) \le{}& \gamma\sqrt{d_v d_u}\Big(\frac{m}{|\mathsf{E}|}\mu-\frac{\tau(v,u)}{4c_2|\mathsf{E}|}\mu\Big)+\frac{\gamma\mu}{2c_2\lambda_1}\Big(\sqrt{\frac{d_v}{d_u}}+\sqrt{\frac{d_u}{d_v}}\Big)\\
&+\frac{\gamma\mu}{c_2\lambda_1}|1-c_2\lambda^*|^{r-1}+2\frac{\gamma c^{(2)}}{c_2}.
\end{aligned}
$$

We can now rearrange the terms and obtain

$$
\begin{aligned}
m \ge{}& \frac{\tau(v,u)}{4c_2}+\frac{|\mathsf{E}|}{\sqrt{d_v d_u}}\Big(\frac{\mathsf{mix}_{y_\mathsf{G}}(v,u)}{\gamma\mu}-\frac{1}{2c_2\lambda_1}\Big(\sqrt{\frac{d_v}{d_u}}+\sqrt{\frac{d_u}{d_v}}\Big)-\frac{1}{c_2\lambda_1}|1-c_2\lambda^*|^{r-1}-2\frac{c^{(2)}}{c_2}\frac{1}{\mu}\Big)\\
\ge{}& \frac{\tau(v,u)}{4c_2}+\frac{|\mathsf{E}|}{\sqrt{d_v d_u}}\Big(\frac{\mathsf{mix}_{y_\mathsf{G}}(v,u)}{\gamma\mu}-\frac{1}{2c_2\lambda_1}(2\gamma)-\frac{1}{c_2\lambda_1}|1-c_2\lambda^*|^{r-1}-2\frac{c^{(2)}}{c_2}\frac{1}{\mu}\Big)\\
\ge{}& \frac{\tau(v,u)}{4c_2}+\frac{|\mathsf{E}|}{\sqrt{d_v d_u}}\Big(\frac{\mathsf{mix}_{y_\mathsf{G}}(v,u)}{\gamma\mu}-\frac{1}{c_2}\Big(\frac{\gamma+|1-c_2\lambda^*|^{r-1}}{\lambda_1}+2\frac{c^{(2)}}{\mu}\Big)\Big),
\end{aligned}
$$

which completes the proof. $\qquad\square$

We note that the case of $\mathbf{A} = \mathbf{A}_{\mathrm{rw}}$ follows easily since one can adapt the previous argument exactly as in the proof of Theorem D.2, which lead to the same bounds once we replace $\gamma$ with $\gamma' = d_{\max}/d_{\min}$.

First, we note that the bounds again simplify further and become sharper if the message-functions $\psi$ in (2) are linear.

**Corollary D.4.** *If the assumptions of Theorem 4.4 are satisfied, and the message-functions $\psi$ are linear – as for GCN, GIN, GraphSAGE – then*

$$m \geq \frac{\tau(v,u)}{4c_2} + \frac{|\mathsf{E}|}{\sqrt{d_v d_u}} \Big( \frac{\mathsf{mix}_{y_\mathsf{G}}(v,u)}{\gamma} - \frac{1}{c_2 \lambda_1} \big( \gamma + |1 - c_2 \lambda^*|^{r-1} \big) \Big).$$

*In fact, if the graph is regular with degree d, then*

$$m \geq \frac{\tau(v,u)}{4c_2} + \frac{|\mathsf{E}|}{d} \Big( \mathsf{mix}_{y_\mathsf{G}}(v,u) - \frac{1}{c_2 \lambda_1} \big( 1 + |1 - c_2 \lambda^*|^{r-1} \big) \Big).$$

### D.4 The case of the unnormalized adjacency matrix

In this Section we extend the analysis on the depth required to induce mixing, to the case of the unnormalized adjacency matrix $\mathbf{A}$. When $\mathbf{A} = \mathbf{A}$, the aggregation in (2) is simply a sum over the neighbours, a case that covers the classical GIN-architecture. In this way, the messages are no longer scaled down by the degree of (either) the endpoints of the edge, which means that, *in principle*, the whole GNN architecture is more sensitive but independent of where we are in the graph. First, we generalize Theorem 4.4 to this setting. We note that the same conclusions hold, provided that the maximal operator norm of the weights is smaller than the maximal degree $d_{\max}$; this is not surprising, since it accounts for the lack of the normalization of the messages.

**Corollary D.5.** *Consider an MPNN as in (2) with $\mathbf{A} = \mathbf{A}$. If $\omega/(\mathsf{w}d_{\max}) + c_1 + c_2 \leq 1$ and $\mathsf{w}d_{\max} \leq 1$, then the minimal depth $m$ satisfies the same lower bound as in Theorem 4.4 with $\gamma = 1$.*

*Proof.* First, we note that in this case

$$\mathsf{S}_{ij} \leq \frac{\omega}{\mathsf{w}} \delta_{ij} + c_1 d_{\max} \delta_{ij} + c_2 A_{ij} \leq d_{\max} \Big( \alpha \delta_{ij} + c_2 (\mathbf{A}_{\mathrm{sym}})_{ij} \Big),$$

where $\alpha = \omega/(\mathsf{w}d_{\max}) + c_1$. In particular, we find that

$$(\mathbf{S}^k \mathbf{1})_i \leq \sum_{p=0}^{k} \binom{k}{p} \alpha^{k-p} c_2^p (\mathbf{A}^p \mathbf{1})_i \leq (d_{\max})^k (\alpha + c_2)^k.$$

Accordingly, we can bound the first sum in (7) as

$$\Big( \sum_{k=0}^{m-1} \mathsf{w}^{2m-k} (d_{\max})^k (\alpha + c_2)^k (d_{\max})^{2(m-k)} \big( \alpha \mathbf{I} + c_2 \mathbf{A}_{\mathrm{sym}} \big)^{2(m-k)} \Big)_{vu} \leq \Big( \sum_{k=0}^{m-1} \mathbf{Z}^{2(m-k)} \Big)_{vu},$$

where we have used the assumptions $\alpha + c_2 \leq 1$, and $\mathsf{w}d_{\max} \leq 1$, and the definition $\mathbf{Z} := \mathbf{I} - c_2 \mathbf{\Delta}$ in Theorem D.2. Since this term is the same one entering the argument in the proof of Theorem 4.4 (once we set $\gamma = 1$) we can proceed in the same way to estimate it. A similar argument works for the sum of the $\mathbf{Q}_k$ terms, which, thanks to our assumptions, can still be bounded as in (31) with $\gamma = 1$ so that we can finally simply copy the proof of Theorem 4.4. $\qquad \square$

**A relative measurement for $\widetilde{\mathsf{OSQ}}$.** To account for the fact that different message-passing matrices $\mathbf{A}$ may lead to inherently quite distinct scales (think of the case where the aggregation is a mean vs when it is a sum), one could modify the over-squashing characterization in Definition 4.2 as follows:

**Definition D.6.** *Given an MPNN with capacity* $(m, \mathsf{w})$, *we define the* **relative over-squashing** *of nodes* $v, u$ *as*

$$\widetilde{\mathsf{OSQ}}_{v,u}^{\mathsf{rel}}(m, \mathsf{w}) := \left( \frac{\sum_{k=0}^{m-1} \mathsf{w}^{2m-k-1} \Big( \mathsf{w}(\mathbf{S}^{m-k})^{\top} \mathrm{diag}(\mathbf{1}^{\top}\mathbf{S}^{k})\mathbf{S}^{m-k} + c^{(2)}\mathbf{Q}_k \Big)_{vu}}{\max_{i,j \in \mathsf{V}} \sum_{k=0}^{m-1} \mathsf{w}^{2m-k-1} \Big( \mathsf{w}(\mathbf{S}^{m-k})^{\top} \mathrm{diag}(\mathbf{1}^{\top}\mathbf{S}^{k})\mathbf{S}^{m-k} + c^{(2)}\mathbf{Q}_k \Big)_{ij}} \right)^{-1}.$$

The normalization proposed here is similar to the idea of relative score introduced in (Xu et al., 2018). This way, a larger scale induced by a certain choice of the message-passing matrix $\mathbf{A}$, is naturally accounted for by the relative measurement. In particular, the relative over-squashing is now quantifying the maximal mixing among a certain pair of nodes $v, u$ compared to the maximal mixing that the same MPNN over the same graph can generate among *any* pair of nodes. In our theoretical development in Section 4 we have decided to rely on the absolute measurement since our analysis depends on the derivation of the maximal mixing induced by an MPNN (i.e. upper bounds) which translate into necessary criteria for an MPNN to generate a given level of mixing. In principle, to deal with relative measurements, one would also need some form of lower bound on the maximal mixing and hence address also whether the conditions provided are indeed sufficient. We reserve a thorough investigation of this angle to future work.

# E  The case of node-level tasks

In this Section we discuss how one can extend our analysis to node-level tasks and further comment on the novelty of our approach compared to existing results in (Black et al., 2023; Di Giovanni et al., 2023). First, we emphasize that the analysis on the Jacobian of node features carried over in Black et al. (2023); Di Giovanni et al. (2023) cannot be extended to graph-level functions and that in fact, *our notion of mixing is needed* to assess how two different node-features are communicating when the target is a graph-level function.

From now on, let us consider the case where the function we need to learn is $\mathbf{Y} : \mathbb{R}^{n \times d} \to \mathbb{R}^{n \times d}$, and as usual we assume it to be equivariant with respect to permutations of the nodes. A natural attempt to connect the results in Black et al. (2023); Di Giovanni et al. (2023) and the expressivity of the MPNNs – in the spirit of our Section 3 – could be to characterize the ***first-order interactions*** (or mixing of order 1) of the features associated with nodes $v, u$ with respect to the underlying node-level task $\mathbf{Y}$ as

$$\mathsf{mix}_{\mathbf{Y}}^{(1)}(v, u) = \max_{\mathbf{X}} \max_{1 \le \alpha, \beta \le d} \left| \frac{\partial (\mathbf{Y}(\mathbf{X}))_v^{\alpha}}{\partial x_u^{\beta}} \right|,$$

where $(\mathbf{Y}(\mathbf{X}))_v \in \mathbb{R}^d$ is the value of the node-level map at $v$. Accordingly, one can then use Theorem C.1 to derive upper bounds on the maximal first-order interactions that MPNNs (2) can induce among nodes. As a consequence of this approach, we would still find that MPNNs struggle to learn functions with large $\mathsf{mix}_{\mathbf{Y}}^{(1)}(v, u)$ if nodes $v, u$ have large commute time. In particular, in light of Theorem C.1, we can extend the measure of over-squashing to the case of first-order interactions for node-level tasks. Once again, below we tacitly assume that the non-linear activation $\sigma$ satisfies $|\sigma'| \le 1$, although it is straightforward to extend the formulation to the general case.

**Definition E.1.** *Given an MPNN as in* (2) *with capacity* $(m, \mathsf{w})$, *we define the* **first-order over-squashing** *of* $v, u$ *as*

$$\mathsf{OSQ}_{v,u}^{(1)}(m, \mathsf{w}) := \left( \mathsf{mix}_{\mathbf{Y}}^{(1)}(v, u) \right)^{-1}.$$

As for the case of graph-level tasks, we can then study a proxy (lower bound) for the node-level over-squashing of order 1 by:

**Definition E.2.** *Given an MPNN as in* (2) *with capacity* $(m, \mathsf{w})$ *and* $\mathbf{S}$ *defined in* (5), *we approximate the* **first-order over-squashing** *of* $v, u$ *as*

$$\widetilde{\mathsf{OSQ}}_{v,u}^{(1)}(m, \mathsf{w}) := \left( (c_{\sigma}\mathsf{w})^m (\mathbf{S}^m)_{vu} \right)^{-1}.$$

It follows then from Theorem C.1, that a necessary condition for an MPNN to learn a node-level function $\mathbf{Y}$ with first-order mixing $\mathsf{mix}_{\mathbf{Y}}^{(1)}(v, u)$ is

$$\widetilde{\mathsf{OSQ}}_{v,u}^{(1)}(m, \mathsf{w}) < \left( \mathsf{mix}_{\mathbf{Y}}^{(1)}(v, u) \right)^{-1}.$$

It is straightforward to argue as in Theorem 4.4 and (Black et al., 2023) for example, to derive that nodes at higher effective resistance will incur higher first-order over-squashing. Accordingly:

*An MPNN as in (2) with bounded capacity, cannot learn node-level functions with high first-order interactions among nodes $v, u$ with high effective resistance.*

**Building a hierarchy of measures.** Although first-order derivatives might be enough to capture some form of over-squashing for node-level tasks, even in this scenario we can study the pairwise mixing induced *at a specific node*, and hence consider the curvature (or Hessian) of the node-level function $\mathbf{Y}$ – which is more expressive than the first-order Jacobian. Accordingly, for a node-level function $\mathbf{Y} : \mathbb{R}^{n \times d} \to \mathbb{R}^{n \times d}$, we say that it has **second-order interactions** (or mixing of order 2) $\mathsf{mix}_{\mathbf{Y}}^{(2)}(i, v, u)$ of the features associated with nodes $v, u$ at a given node $i$ when

$$\mathsf{mix}_{\mathbf{Y}}^{(2)}(i, v, u) = \max_{\mathbf{X}} \left\| \frac{\partial^2 (\mathbf{Y}(\mathbf{X}))_i}{\partial \mathbf{x}_u \partial \mathbf{x}_v} \right\|.$$

We can then restate Theorem C.2 as follows – we let $\mathbf{Y}^{(m)}$ be the node-level function computed by an MPNN after $m$ layers.

**Corollary E.3.** *Given MPNNs as in (2), let $\sigma$ and $\psi^{(t)}$ be $\mathcal{C}^2$ functions and assume $|\sigma'|, |\sigma''| \leq c_\sigma$, $\|\mathbf{\Omega}^{(t)}\| \leq \omega$, $\|\mathbf{W}^{(t)}\| \leq \mathsf{w}$, $\|\nabla_1 \psi^{(t)}\| \leq c_1$, $\|\nabla_2 \psi^{(t)}\| \leq c_2$, $\|\nabla^2 \psi^{(t)}\| \leq c^{(2)}$. Let $\mathbf{S} \in \mathbb{R}^{n \times n}$ be defined as in (5). Given nodes $i, v, u \in \mathsf{V}$, if $\mathsf{P}_{(vu)}^{(\ell)} \in \mathbb{R}^n$ is as in (15) and $m$ is the number of layers, then the maximal mixing of order 2 of the MPNN at node $i$ satisfies*

$$\mathsf{mix}_{\mathbf{Y}^{(m)}}^{(2)}(i, v, u) \leq \sum_{k=0}^{m-1} \sum_{j \in \mathsf{V}} (c_\sigma \mathsf{w})^{2m-k-1} \mathsf{w}(\mathbf{S}^{m-k})_{jv} (\mathbf{S}^k)_{ij} (\mathbf{S}^{m-k})_{ju}$$

$$+ c^{(2)} \sum_{\ell=0}^{m-1} (c_\sigma \mathsf{w})^{m+\ell} (\mathbf{S}^{m-1-\ell} \mathsf{P}_{(vu)}^{(\ell)})_i. \tag{41}$$

Similarly to Definition 4.2, we can use the maximal mixing (at the node-level) to characterize the over-squashing of order two at a specific node as follows: as usual, for simplicity we assume that $c_\sigma = 1$.

**Definition E.4.** *Given an MPNN as in (2) with capacity $(m, \mathsf{w})$ and $\mathbf{S}$ defined in (5), we approximate the* **second-order over-squashing** *of $v, u$ at node $i$ as*

$$\widetilde{\mathsf{OSQ}}_{i,v,u}^{(2)}(m, \mathsf{w}) := \Big( \sum_{k=0}^{m-1} \sum_{j \in \mathsf{V}} \mathsf{w}^{2m-k} (\mathbf{S}^{m-k})_{jv} (\mathbf{S}^k)_{ij} (\mathbf{S}^{m-k})_{ju}$$

$$+ c^{(2)} \sum_{\ell=0}^{m-1} \mathsf{w}^{m+\ell} (\mathbf{S}^{m-1-\ell} \mathsf{P}_{(vu)}^{(\ell)})_i \Big)^{-1}.$$

It is then straightforward to extend our theoretical analysis to derive how $\widetilde{\mathsf{OSQ}}^{(2)}$ prevents MPNNs from learning node-level functions with high-mixing at some specific node $i$ of features associated with nodes $v, u$ at large commute time. To support our claim, consider the setting in Theorem 4.4 and hence let $\mathbf{A} = \mathbf{A}_{\mathrm{sym}} = \mathbf{D}^{-1/2} \mathbf{A} \mathbf{D}^{-1/2}$. Under the same assumptions of Theorem 4.4, we find

$$(\mathbf{S}^k)_{ij} \leq 1.$$

We can then simply copy the proof of Theorem 4.4 once we set $\gamma = 1$ and extend its conclusions as follows:

**Corollary E.5.** *Consider an MPNN as in* (2) *and let the assumptions of Theorem 4.4 hold. If the MPNN generates second-order mixing* $\mathsf{mix}_{\mathbf{Y}}^{(2)}(i, v, u)$ *at node i, with respect to the features associated with nodes* $v, u$, *then the number of layers m satisfy:*

$$m \geq \frac{\tau(v,u)}{4c_2} + \frac{|\mathsf{E}|}{\sqrt{d_v d_u}}\Big(\frac{\mathsf{mix}_{y_{\mathsf{G}}}(v,u)}{\mu} - \frac{1}{c_2}\Big(\frac{1 + |1 - c_2\lambda^*|^{r-1}}{\lambda_1} + 2\frac{c^{(2)}}{\mu}\Big)\Big),$$

*where* $\mu = 1 + 4c^{(2)}$.

Accordingly, in this Section we have adapted our results from graph-level tasks to node-level tasks and proved that:

**The message of Section E.** *An MPNN of bounded capacity* $(m, \mathsf{w})$, *cannot learn node-level functions that, at some node i, induce high (first order or second order) mixing of features associated with nodes* $v, u$ *whose commute time is large.*

## F   Additional details of experiments and further ablations 5

### F.1   Computing the commute time

The commute time $\tau$ between two nodes $u, v \in \mathsf{V}$ on a graph $\mathsf{G}$ can be efficiently computed via the effective resistance $\mathsf{R}$, with $\tau(u,v) = 2|\mathsf{E}|\mathsf{R}(u,v)$. In order to compute the effective resistance $\mathsf{R}$, we introduce the (non-normalized) Laplacian matrix $\mathbf{L} = \mathbf{D} - \mathbf{A}$, where $\mathbf{D}$ is the degree matrix. The effective resistance can then be computed by

$$\mathsf{R}(u,v) = \Gamma_{uu} + \Gamma_{vv} - 2\Gamma_{uv},$$

where $\mathbf{\Gamma}$ is the the Moore-Penrose inverse of

$$\mathbf{L} + \frac{1}{|V|}\mathbf{1}_{|V| \times |V|},$$

with $\mathbf{1}_{|V| \times |V|} \in \mathbb{R}^{|V| \times |V|}$ being a matrix with all entries set to one.

### F.2   On the training error

In this section, we report the training error of the MPNNs trained in section 5. Fig. 5 shows the training MAE corresponding to the experiment in section 5.1, while Fig. 6 shows the training MAE corresponding to section 5.2. We can see that in both cases, the training MAE exhibits the same qualitative behavior as the reported test MAE in the main paper, i.e., the training MAE increases for increasing levels of commute time $\tau$, while it decreases for increasing number of MPNN layers, which further validates our claim that *over-squashing hinders the expressive power of MPNNs.*

### F.3   On the over-squashing measure

In this section, we examine how the over-squashing measure $\widetilde{\mathsf{OSQ}}$ (as of Definition 4.2) depends on the commute time $\tau$ as well as on the depth of the underlying MPNN. To this end, we follow the experimental setup of section 5.1 and 5.2, but instead of training the models and presenting their performance in terms of the test MAE, we compute $\widetilde{\mathsf{OSQ}}$ of the underlying models. We can see in Fig. 7 that $\widetilde{\mathsf{OSQ}}$ increases for increasing values of the $\alpha$-quantile of the $\tau$-distribution for all MPNNs considered here. Moreover, we can see in Fig. 8 that $\widetilde{\mathsf{OSQ}}$ decreases for increasing number of layers for all considered models.

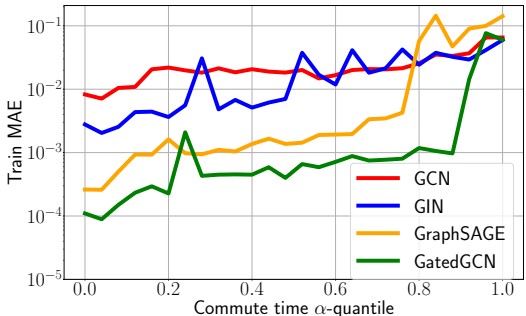

Figure 5: Train MAE of GCN, GIN, GraphSAGE, and GatedGCN on synthetic ZINC, where the commute time of the underlying mixing is varied, while the MPNN architecture is fixed (e.g., depth, number of parameters), i.e., mixing according to increasing values of the $\alpha$-quantile of the $\tau$-distribution over the ZINC graphs.

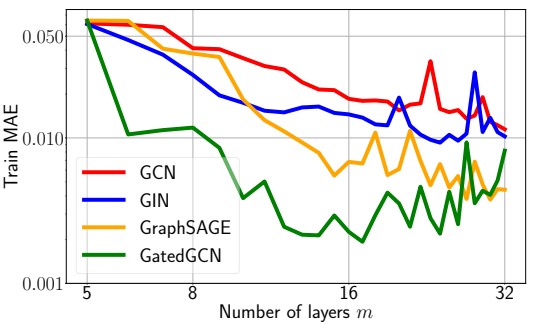

Figure 6: Train MAE of GCN, GIN, GraphSAGE, and GatedGCN on synthetic ZINC, where the commute time is fixed to be high (i.e., at the level of the 0.8-quantile), while only the depth of the underlying MPNN is varied between 5 and 32 (all other architectural components are fixed).

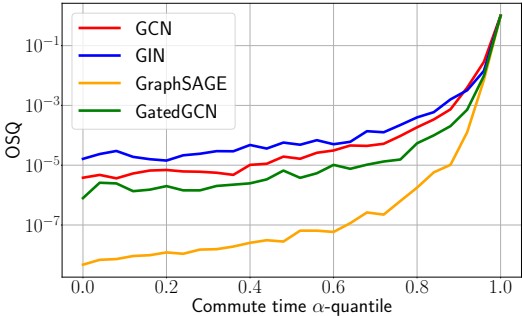

Figure 7: $\widetilde{\text{OSQ}}$ (Definition 4.2) of GCN, GIN, GraphSAGE, and GatedGCN on synthetic ZINC, where the commute time of the underlying mixing is varied, while the MPNN architecture is fixed (e.g., depth, number of parameters), i.e., mixing according to increasing values of the $\alpha$-quantile of the $\tau$-distribution over the ZINC graphs.

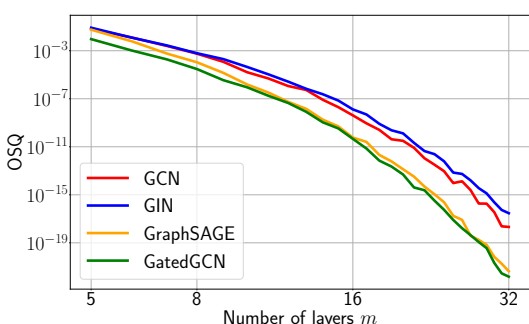

Figure 8: $\widetilde{\text{OSQ}}$ (Definition 4.2) of GCN, GIN, GraphSAGE, and GatedGCN on synthetic ZINC, where the commute time is fixed to be high (i.e., at the level of the 0.8-quantile), while only the depth of the underlying MPNN is varied between 5 and 32 (all other architectural components are fixed).

