# OpenReview forum: "How does over-squashing affect the power of GNNs?"
_TMLR — Accepted by TMLR_

### Review · Reviewer_AJhv · 2023-11-06

**Summary Of Contributions:**

Summary:

a) The authors re-define the over-squashing phenomena previous defined in Topping et al. 2022, Di Giovanni et al. 2023, in terms of the "mixing" induced by the MPNN. The ``mixing" here is defined, assuming the output/labeling function is a smooth function of the input, node features. Precisely it defined as the (maximum) magnitude of the mixed derivatives (w.r.t two different nodes) of this labeling function. Once this definition is made then over-squashing between two nodes is defined as the inverse of the mixing between the nodes. More importantly, the provide an estimate of this that depends on the norms of the (learnable) weight matrices and crucially also the graph topology and parameters such as the commute time between two nodes, max/min degrees, the eigenvalues of the Laplacian etc.

b)Based on the above estimates, they provide some necessary conditions on the the depth (with fixed weight) and the norm (with fixed depth) in order to achieve a certain mixing value.

c)They also are able to explain certain phenomena observed in other prior-works such as under-reaching, in terms of mixing.

d) They have controlled experiments explaining the effects and show that these experiments reflect the theoretically predicted patterns.

**Audience:**

Yes

**Broader Impact Concerns:**

It is predominantly a theoretical paper. There are no ethical implications as far as I can see.

**Claims And Evidence:**

Yes

**Requested Changes:**

Minor:

a)Can the formulae for over-squashing, mixing etc be left intact with their respective norms instead of the bounds. I think it might help to see the general dependence on norms easily.

b) Some notations such as double-brackets and the MAX operator should be explained for better readability and why is the function.

c) Corollary 4.5 seems quite informal as stated here. How high is "high" and how large is "large"?

Major:

a) Di Giovanni et al. 2023 show that increasing depth may not prevent over-squashing (according to their definition) and leads to vanishing gradients. I think this is important to address here in this paper, which uses a different definition of over-squashing. But then can one increase depth indefinitely without any issues?

b) Does mixing/over-squashing dependence on the initial node-features, any dependence on $X$, or the dimension of the node features? I think this has to be clarified slightly better.

c)The requirement of twice-differentiable output function (w.r.t to the node features) maybe quite strong in general. It would be great if the authors could have some discussion of how to overcome and estimate the mixing outside this setting?

**Strengths And Weaknesses:**

Strengths:

a) They have a reasonable justification for the way they define mixing and as a result over-squashing, as required by the underlying task, which is absent from prior works.

b) Their measure seemingly captures different phenomena of under-reaching, redundancy of messages that were observed in prior works such as Barceló et al. 2019 and Chen et al.2022.

Weaknesses:

A lot of the main message of the paper is not entirely new/unknown in relation to the prior works such as Di Giovanni et al.2023, Black et al.2023 etc, especially in relation with the commute times of the graph.

---

> ### Author Response · Authors · 2023-11-07
> **Rebuttal**
>
> We thank the reviewer for their comments and feedback. We address your concerns in full here and point out to modifications of the manuscript:
>
> **Regarding novelty compared to previous work**
>
> First, we kindly point out to Section 6, where we have commented on the differences between our submission and previous works on over-squashing extensively. In particular, we emphasize that _no previous work has formulated over-squashing as a limitation of the expressivity of a model to capture the mixing of the underlying task_, which we believe to be an important, novel contribution.
>
> We also refer to Section E in the Appendix, where we have generalized our theoretical definitions and derivations to node-level tasks and explained how existing works of Topping et al. (2022), Di Giovanni at al. (2023), and Black et al. (2023), can in fact be regarded as a first-order approximation of the over-squashing phenomenon, which we argue may not be sufficient to capture the notion of mixing -- which instead is related to product-type of interactions and hence derivatives of order higher than one. _By studying the mixing and hence second-order derivatives, our definitions also work for graph-level tasks while the formulae derived in previous works would not easily extend to such scenarios_.
>
> Here we would like to comment further on how our work stands out for its connections to *commute time*;
>
> - For what concerns the work of Di Giovanni et al. (2023), they manage to show a connection to commute time only in a simplified setting: they assume that the derivatives of the non-linear activation are, *in expectation*, the same for each path in the computational graph. This is a strong assumption; in our setting, we manage to derive results that are exact without any additional assumptions.
> - Regarding the work of Black et al. (2023), we highlight some important points: (i) Their main result (Theorem 3) presents a connection between Jacobian sensitivity and *effective resistance* $R$. While the commute time $\tau$ is proportional to $R$, since it satisfies $\tau = 2 | \mathsf{E} | R$, the fact that their bound is in terms of $R$ rather than $\tau$ is significant: in fact, no particularly meaningful lower bound in terms of the number of layers can be derived from their Theorem 3 since the effective resistance is always smaller or equal than the pairwise shortest-walk. In fact, any lower bound on the number of layers in terms of the effective resistance would actually be weaker than saying that the number of layers has to be at least as large as the pairwise distance (which is the under-reaching case). In our setting instead, we manage to derive a lower bound in terms of the commute time, which is larger (typically much larger) than the pairwise distance, hence leading to a **stronger** result. (ii) Their analysis relies on Lemma 3.2 which is actually not optimal, note for example that if the GNN was linear, then only the term with higher power should appear in the sum. This is partly due to the analysis of first-order derivatives not being an optimal approach, as we have argued in Section E of the appendix.
> - Finally, as commented in Section 6 of the submission, previous works have failed to derive results for general MPNNs as in equation (2) of our work -- for example neither Di Giovanni et al. (2023) nor Black et al (2023) results apply to architectures like GatedGCN.
>
> We hope this clarifies why our contributions are novel, and the connections to commute time are deeper and more significant than existing works.
>
> **Regarding requested changes**
>
> _Minor_
>
> a) We are not sure we get the point, could you please elaborate more on that?
>
> b) Thanks for the feedback. We have added a clarification on the double bracket notation. Regarding the MAX, MEAN, SUM note that we have specified how these can be taken as instances of the READ map in Equation (3).
>
> c) Thanks for your observation. We have rewritten Corollary 4.5 more explicitly to emphasize how it follows from Theorem 4.4 We have also slightly rearranged the comments after Theorem 4.4. to improve the flow; hope this has improved the readability.
>
> Please see below for comments on Major requested changes.

---

> ### Author Response · Authors · 2023-11-07
> **Rebuttal part II**
>
> _Major_
>
>
> a) Thank you for the question. Two crucial observations:
>
> - (1) Our results are not in contrast with Di Giovanni et al. (2023). Di Giovanni et al (2023) have shown that MPNNs can suffer from vanishing gradients in the limit of infinite layers, and hence they may lose sensitivity altogether. Our theoretical results do not claim that having a large number of layers is always going to be optimal for the task. Instead, our analysis shows how the number of layers depends, precisely, on the mixing required by the task and on the graph-topology (commute time). However, we have argued in our paper that such lower bound may translate into an *impossibility statement in practice*, since large depths come with issues such as scalability, over-smoothing, or indeed vanishing gradients. Put differently, as stated in the new Corollary 4.5, if the depth does not satisfy our proposed lower bound, **then the MPNN cannot learn functions with mixing $\mathsf{mix}\_{y\_\mathsf{G}}(v,u)$ independent of other phenomena such as vanishing gradients**. When this lower bound is satisfied, then if $m$ is too large, other issues can arise, such as vanishing gradients.  We have added a comment below Theorem 4.4 and a reference to Di Giovanni et al. (2023) to point out that our results corroborate that over-squashing limits the expressivity of the MPNN since large depths can also lead to vanishing gradients.
> - (2) The vanishing-gradient result of Di Giovanni et al. (2023) does not apply to the whole class of models analysed in Theorem 4.4. In fact, the results of vanishing gradients from Di Giovanni et al. (2023) would not apply, for example, to MPNNs with $\omega = 0, c_1 = 0, c_2 = 1, \mathsf{w} = 1$ (this represents the boundary case in Theorem 4.2 of Di Giovanni et al. (2023)), while our Theorem 4.4 **still holds for such cases** where we do not know, a priori, if gradients are going to vanish, stay finite, or explode. We hope this addresses your question.
>
> b) Thanks for the questions. Yes the mixing, and hence our measure of over-squashing, depend on the features $\mathbf{X}$. In fact, the mixing is computed using the second-order derivatives of the task with respect to input features at different nodes, and hence it would be a (generally nonlinear) function of the features; concretely, say that the task is a function $y(x_v, x_u) = \phi(x_v^\top x_u)$, with $\phi$ non-linear, then the second-order derivatives of $y$ are themselves functions of $x_v$ and $x_u$. Having said that, our theoretical upper bounds on the mixing in Theorem 3.2 is a number, and not a function anymore, depending on properties of the network such as maximal norm of the weights and maximal derivatives; this way, we can control the true mixing (i.e. true over-squashing) which is a function, in terms of a maximum which is a number. Hope this addresses your question.
>
> c) Thanks for the feedback. We disagree that the requirement of twice differentiability is strong. We have already commented in the manuscript that the majority of non-linear activations used in practice are actually smooth, and even those that are not, such as ReLU, can be replaced by smooth versions like the GELU function, with no theoretical or practical downside. Besides, in many applications such as Neural PDEs or Force-Field models, the functions we need to approximate are *smooth*, meaning that the network itself has to include smooth activations and cannot rely on functions like ReLU. Having said that, it should in principle be possible to extend the analysis by replacing derivatives with finite differences so to argue with approximation in $C^0$ rather than in $C^2$. However, we believe that this is outside the scope of the current submission and hope we have convinced you why we believe having theoretical derivations that apply to activations that are, at least, twice differentiable, is not a strong limitation.

---

### Review · Reviewer_gw29 · 2023-12-13

**Summary Of Contributions:**

This paper studies over-squashing for message-passing GNNs (MPNN). The authors introduce a notion of mixing for a function of node features, which is an upper bound of the Hessian. In Section 3, they derive an upper bound for the mixing of MPNN as a function of the bounds on the weights, gradients of the aggregating functions, and activation functions. Then, Section 4 defines a quantitative notion of over-squashing (OSQ) as the inverse of the mixing defined above. A direct consequence of their first result is that in order to be able to approximate a function with a given mixing, the OSQ of the MPNN needs to be low enough. The authors derive bounds on the weight norm (Theorem 4.3) and the minimal number of layers (Theorem 4.4) for this condition to be valid. These theoretical results are validated through experiments on synthetic signals on real graphs.

**Audience:**

Yes

**Claims And Evidence:**

No

**Requested Changes:**

It looks like the bounds can be very loose. For example, in Theorem 4.4, deriving a lower bound on the number of layers, it is not clear that this bound is indeed positive. For a GCN, we have a term like $\frac{A}{\gamma} -(\gamma+B)$, so if $\gamma$ is large, this will be negative.
The experiments do not help and, indeed, are in favor of claiming that these bounds are very loose. The mixing is of order one, and the OSQ given in Appendix F.3. varies between $10^{-19}$ and $10^{-3}$, so very far from the targeted mixing. Clearly, it would be much more convincing to have experiments showing the tightness of the bounds. This is not the case here.

**Strengths And Weaknesses:**

Strengths:
- an interesting topic for which getting theoretical results is not easy.
- the paper is well-written and easy to follow

Weakness:
- it is not clear that the bounds derived are indeed tight. In this case, the theoretical results will not explain the expressive power of MPNN.
- the experimental results are weak (see below).

---

> ### Author Response · Authors · 2023-12-28
> **Rebuttal**
>
> We thank the reviewer for their feedback and apologize for the late response but the timing was not optimal.
>
> We structure the response in three parts:
>
> **1. On Theorem 4.4**
>
> To make things more concrete, consider the case of GCN. The lower bound in Theorem 4.4. then simplifies as:
>
> $$m \geq  \frac{\tau(v,u)}{4} + \frac{\lvert\mathsf{E}\rvert}{\sqrt{d_v d_u}}\Big(\frac{\mathsf{mix}\_{y\_\mathsf{G}}(v,u)}{\gamma} - \Big(\frac{\gamma + |1 - \lambda^\ast|^{r-1}}{\lambda_1}\Big)\Big).$$
>
> The way this lower bound should be interpreted is: fix a message-passing neural network and a graph $\mathsf{G}$, we can then find (infinitely many) tasks (i.e. graph-functions) $y\_\mathsf{G}$ whose mixing between $v,u$ is large enough, such that
>
> $$  m \geq  \frac{\tau(v,u)}{4}. $$
>
> Accordingly, **_for tasks involving sufficiently strong mixing between pairs of nodes $v,u$, the minimal number of layers required to match such mixing scales with the commute time between the nodes (and not, for example, with their shortest-walk distance)_**.
>
> In fact, *how large* the mixing of a task is, depends on properties of the graph-topology. For example, for regular graphs (which include those that are not distinguishable by the 1-WL test), one can simplify further the lower bound since $\gamma = 1.$
>
> Therefore, Theorem 4.4. does not say that for any graph, and for any task, the number of layers would have to increase with the commute time between the nodes. Instead, it shows that for tasks that have a large enough mixing, depending on the topology, the number of layers has to scale with the commute time.
>
> *This is the first theoretical characterization of this sort; no previous work has managed to relate properties of the underlying task (its mixing), of the architecture (its depth), and of the topology (commute time) so rigorously, and we hope the reviewer can appreciate that.*
>
> **2. On the experimental evaluation**
>
> First, we would like to point out that the plots reported in Appendix F.3 are rescaled by dividing the oversquashing measure by the maximal value. More importantly, we have not enforced any bound on the spectral norm of the weights. It is possible to attain curve with smaller variations of orders of magnitude, by enforcing some regularization at the level of the spectral norm of the weights. Nonetheless, the bounds will not be of order $O(1)$ when the commute time between the source and target nodes approaches large percentile. This is to be expected though, since, as argued above, we are bounding the action of the network through Lipschitz constants of the layer functions.
>
> We believe that our experimental results actually validate our theoretical findings, fully confirming that our analysis captures the qualitative behaviour depicted by the theorems and the limitations on the expressive power.
>
> - For the experiments in Figure 3, our theory fully justifies why and how the performance deteriorates when we increase the commute time between source and target, **even though the number of layers is sufficient to avoid underreaching**. In fact, note that in Fig. 5 of the Appendix we have also shown how the same plots can be obtained for the **training error**.
>
> - For the experiment in Figure 4, our theory fully justifies why increasing the number of layers **significantly past** the one required to avoid underreaching, is quite beneficial. *This shows that while Theorem 4.4. can not be tight for certain topologies, it certainly captures the right qualitative behaviour, where the depth of the model might often be much larger than the one given by the shortest-walk distance.*
>
> - For the experiment in Table 1, our theory justifies how the performance correlates with the mixing---note that the same experiments using a FNN without a graph do not exhibit variations. Accordingly, our theory does predict limitations to the expressive power of MPNNs.
>
> _If you believe that any of our experimental results could be justified **without our theoretical analysis**, we would be happy to comment on it further._
>
> **3. On achieving quantitative and practical bounds for "Lipschitz"-style analysis**
>
> One of the main reasons why the bounds are not tight, is that they depend on the spectral norm of the weight matrices entering each layer of the GNN. However, this is natural whenever one analyzes deep nonlinear neural networks and hence has to rely on Lipschitz constants of the network. A similar observation for example, would apply to generalization bounds that depend on the spectral norm of the weight matrices.
>
> As the reviewer has pointed out, deriving theoretical results of this kind, especially under very minimal assumptions on the GNN architecture, is not easy and we believe that our bounds characterize qualitatively, rather than quantitatively, the oversquashing phenomenon and its impact on expressive power.

---

### Review · Reviewer_7rF7 · 2023-12-14

**Summary Of Contributions:**

Authors provide a rigorous analysis to determine which function classes of node features can be learned by an Message Passing Neural Networks of a given capacity using a the level of pairwise interactions between nodes that Message Passing Neural Networks allow for.
Theoretical and practical results supports the proposed approach.

**Audience:**

Yes

**Broader Impact Concerns:**

No concerns.

**Claims And Evidence:**

Yes

**Requested Changes:**

I think that the paper should be improved by
- increase the experimental results
- simplify the paper by moving the more technical parts in the appendix, better explain in plain english the results
- add some toy examples to provide more insights into the theoretical results

**Strengths And Weaknesses:**

Strengths
- novelty
- scientific rigour
- theoretical results
Weaknesses
- experimental results are quite limited
- paper is very hard to read and understand

---

> ### Author Response · Authors · 2023-12-14
> **Rebuttal**
>
> We thank the reviewer for their feedback and for appreciating the novelty and theoretical results.
>
> **About making the paper more accessible**
>
> Thank you for the feedback.
>
> First, we kindly point to existing parts of the submission where we made an effort to explain the more theoretical sections in more accessible format. In particular, we refer to the discussion at the beginning of Section 3 to motivate the notion of mixing, the paragraph on the *strategy* on page 7, and finally Corollary 4.5 - where we reformulated the content of the Theorem 4.4 in much simpler terms.
>
> **_To further enhance the accessibility of our submission, we have also added_** boxed summary paragraphs at the end of Section 3 and Section 4, respectively (_please see the revised document_). In these boxes, we have included main messages of the sections and theoretical conclusions from a high-level perspective, without resorting to equations. We hope this addresses your concerns and improves the clarity of our paper.
>
> **About the experimental part of our submission**
>
> We kindly point out to the discussion of our experimental results in Section 5, where we have amply explained how the different ablations fully validate our theoretical results. If there is a specific aspect of these experiments that you find not satisfactory or clear, please let us know and we would do our best to address this point. We believe that our experiments are in fact extensive, since they check different aspects of our theoretical results:
> - the impact of the commute time (i.e. the graph-topology) in the performance of MPNNs - a task depending only on predicting a function of two node features will do much worse if the commute time between these two nodes is large;
> - the impact of the depth - we have shown that increasing the number of layers significantly past the one required to avoid underreaching is actually beneficial, something that resonates with Theorem 4.4 and Corollary 4.5, which show that the number of layers would generally grow with the commute time and not just the distance among nodes;
> - the impact of the mixing of the task - _we have shown that if we fix the nodes but change the task to require more mixing, the performance of the GNNs deteriorates significantly, confirming that our metric of mixing does indeed capture a form of expressive power_.
>
>
> We hope this clarifies your doubts and we are happy to engage in further discussion.

---

> > ### Comment · Reviewer_7rF7 · 2023-12-14
> >
> > I think that more dataset and more architectures should be tested.
> > I agree that the experiments performed are ok but they are too limited.

---

> > > ### Author Response · Authors · 2023-12-28
> > > **Response**
> > >
> > > Thank you for your feedback.
> > >
> > > Considering that this is a theoretical paper that for the *first time*, presents novel results relating the task (mixing), the architecture (norm of the weights and depth), and the topology (commute time), we believe that our experimental evaluation is pertinent. We have thoroughly validated each claim in our theoretical analysis, considering main instances of the Message-Passing class to which the theory applies.
> > >
> > > In fact, to fully validate our theoretical claims we needed to design our own task. For these reasons, we do not believe that adding more datasets or architectures, will improve the message of the paper.

---

### Decision · Action_Editor_VYxU · 2024-02-10

**Recommendation:** Accept as is

**Comment:**

The final feeling of reviewers was mixed, although suggestion to accept the paper were in majority. The main perplexities were concerning the practical application of the theory, as well as  the strength of the experimental assessment proposed by the paper. The decision to accept it is mainly based on the recognition of the theoretical nature of the paper. Although direct practical applications of the theory are far from been clear, the proposed theory constitutes a step towards a better understanding of learning in this type of networks. As such, the work proposed in the paper deserves to be presented to the scientific community as a potential new starting point from where to develop more advanced theories, as well as guidelines for designing more effective architectural features of graph neural network models.

**Audience:**

The paper can be of interest for individuals working on the development of graph neural networks theoretical results, as well as to individuals working on the development of new and more effective graph neural network architectures.

**Claims And Evidence:**

In the context of Graph Neural Networks, the paper proposes a measure of the level of pairwise interactions between nodes of a graph which is used as basis to develop a theory characterizing the function classes of node features that can be learned by a model of a given capacity. The inverse of this measure is considered to be a quantitative proxy for the well-known over-squashing phenomena that hinder learning in graph neural networks.
The contribution is technically sound. I believe that the proposed experimental assessment of the theory is not the strongest one, but it seems to be sufficient to support the main claims of the paper. In fact, although the proposed theory cannot be considered directly applicable to practical applications of graph neural networks, it constitutes a step forward in a better understanding of learning in these type of networks. Authors have discussed limitations of the proposed theory, which provides directions of development for the research community.